# Demystifying the Optimal Performance of Multi-Class Classification

**Minoh Jeong**
Electrical and Computer Engineering
University of Minnesota
Minneapolis, MN 55455
jeong316@umn.edu

**Martina Cardone**
Electrical and Computer Engineering
University of Minnesota
Minneapolis, MN 55455
mcardone@umn.edu

**Alex Dytso**
Qualcomm Flarion Technology, Inc.
Bridgewater, NJ 08807
odytso2@gmail.com

## Abstract

Classification is a fundamental task in science and engineering on which machine learning methods have shown outstanding performances. However, it is challenging to determine whether such methods have achieved the Bayes error rate, that is, the lowest error rate attained by any classifier. This is mainly due to the fact that the Bayes error rate is not known in general and hence, effectively estimating it is paramount. Inspired by the work by Ishida et al. (2023), we propose an estimator for the Bayes error rate of supervised multi-class classification problems. We analyze several theoretical aspects of such estimator, including its consistency, unbiasedness, convergence rate, variance, and robustness. We also propose a denoising method that reduces the noise that potentially corrupts the data labels, and we improve the robustness of the proposed estimator to outliers by incorporating the median-of-means estimator. Our analysis demonstrates the consistency, asymptotic unbiasedness, convergence rate, and robustness of the proposed estimators. Finally, we validate the effectiveness of our theoretical results via experiments both on synthetic data under various noise settings and on real data.

## 1 Introduction

Supervised classification problems are typical tasks in various fields of science and engineering, such as machine learning, statistical signal processing, estimation, and detection. In supervised classification, a dataset consisting of several input feature-label pairs is given. The goal of a supervised classification task is to design effective classifiers, by leveraging the given dataset, to suitably label future input features, or equivalently, to classify future input features into one of the classes.

As the dataset is the only available resource on the data distribution, the performance of a classifier is typically measured by its empirical misclassification rate on the test dataset (which is a subset of the given dataset). The best performance of an existing classifier, that is, the so-called state-of-the-art (SOTA) performance, tends to be the point of reference to measure the improvement of a new classifier. However, there are no guarantees that the SOTA performance is close to the theoretical minimum misclassification rate, namely the Bayes error rate (BER). Thus, comparing the empirical misclassification rate with the SOTA error rate provides only a relative improvement.

Having the knowledge of the BER is essential in theory and practice. The BER indeed provides a fundamental limit on the misclassification rate, which is critical for designing high-performing

37th Conference on Neural Information Processing Systems (NeurIPS 2023).

classifiers. Moreover, one can leverage the BER to assess how good the SOTA performance is with respect to the theoretically optimal error rate. If the SOTA performance is nearly close to the BER, we can avoid wasting time and effort in designing a new classifier. Furthermore, the BER indicates the inherent hardness of a task and hence, it can be seen as a benchmark for comparing the hardness of different tasks [13, 46, 83]. Knowing the BER also brings the opportunity to detect whether test dataset overfitting occurs (which has sporadically happened [3, 51, 59, 91]); this overfitting can be detected since the BER provides the minimum misclassification rate and hence, no classifier will perform strictly better than it. We refer an interested reader to Appendix A for a thorough literature overview on methods to estimate and bound the BER.

In this paper, we investigate the problem of estimating the BER of an $M$-class classification task directly from a dataset, where $M \geq 2$ is arbitrary. Our technique to estimate the BER is different from a plug-in approach that first estimates the distribution from which the data is drawn, and then evaluates the BER. Indeed, our BER estimators, which are proved to be unbiased, consistent and robust to label noise and outliers, do not require the estimation of the data probability density to perform an effective BER estimation. We start by assuming that the data labels are soft and real-valued, approximating the posterior probability of the class. We then relax this assumption on the data labels, and show the applicability of our estimators on one-hot labels and other noisy datasets.

**Contributions.** Our contribution is summarized as follows. Inspired by [46], in Section 3 we first propose a BER estimator for the case of soft data labels, and we show that it benefits from several appealing properties, e.g., it is consistent, unbiased and asymptotically normal. In Section 3, we also propose a methodology, inspired by the median-of-means estimator [65], to make any BER estimator robust. Then, in Section 4 we study the performance of the proposed estimators in scenarios where the soft labels are corrupted by two typical types of noise, i.e., the case of a noise that permutes/shuffles the labels, and the case of additive noise. For the first type of noise, our estimators have similar properties as in the noiseless case. However, for the additive noise case, our proposed estimators are not consistent. Because of this, we propose a denoising method that averages noisy labels associated with the same feature. The corresponding constructed estimator is shown to be consistent. In Section 4, we also showcase that the noisy soft label framework can be used to study the case of one-hot labels, and we provide a denoising method that encompasses the one proposed for the case of additive noise. In Section 5 we validate the effectiveness of the proposed estimators via experiments both on synthetic data under various noise settings (e.g., one-hot labels) and on real data using three different datasets, i.e., CIFAR-10H [4], Fashion-MNIST-H [46] and MovieLens [38]. Finally, in Section 6 we conclude the paper with some discussion on future research directions, which are worth further investigation.

**Notation.** Deterministic scalar quantities are denoted by lowercase letters, scalar random variables are denoted by uppercase letters, vectors are denoted by bold lowercase letters, and random vectors by bold uppercase letters (e.g., $x$, $X$, $\boldsymbol{x}$, $\mathbf{X}$). We let $x_i$ (resp., $(\boldsymbol{x}_k)_i$) indicate the $i$-th value of $\boldsymbol{x}$ (resp., $\boldsymbol{x}_k$). Calligraphic letters $\mathcal{X}$ denote sets, and $|\mathcal{X}|$ is the cardinality of $\mathcal{X}$. $\mathbb{1}\{\mathcal{S}\}$ is the indicator function that yields 1 if $\mathcal{S}$ is true and 0 otherwise. $[M] := \{1, \dots, M\}$. For $\mathbf{X} \in \mathbb{R}^M$, we let $X_{i:M}$ be the $i$-th order statistics [18] of $\mathbf{X}$, i.e., the $i$-th smallest value of $\mathbf{X}$ with $i \in [M]$. Finally, $I_n$ is the identity matrix of dimension $n$, and $\xrightarrow{d}$ (resp., $\overset{d}{=}$) denotes convergence (resp., equality) in distribution.

## 2 Problem formulation

We consider an $M$-class classification task in which an input feature $x$ is classified into a class $c \in \mathcal{C} := [M]$. Our goal is to estimate the minimum misclassification probability, that is the BER. In particular, we seek to estimate the BER based on a dataset $\mathcal{D} = \{(\boldsymbol{x}_i, \boldsymbol{y}_i)\}_{i=1}^n$ that follows an unknown data distribution, i.e., $(\boldsymbol{x}_i, \boldsymbol{y}_i) \overset{i.i.d.}{\sim} P_{\mathbf{X}, \mathbf{Y}}$, where $\mathbf{X} \in \mathcal{X}$ and $\mathbf{Y} \in \mathcal{Y} \subseteq [0, 1]^M$.[1]

We assume that the label data $\boldsymbol{y}$ contains the information about the class $c$ of the input feature $\boldsymbol{x}$, implying that one can retrieve $(\boldsymbol{x}, c)$ from $(\boldsymbol{x}, \boldsymbol{y})$. When a classifier $\phi : \mathcal{X} \to \mathcal{C}$ is employed for a classification task, the corresponding misclassification probability is defined as $\mathcal{E}(\phi) := \Pr(\phi(\mathbf{X}) \neq C)$, where $C$ is the true class for $\mathbf{X}$. The minimum value of this misclassification probability is the so-called BER, denoted by $P_e$, which is formally defined next.

---

[1]We assume that $\mathcal{X}$ is a finite set, but several of our results easily extend to the case when $\mathcal{X}$ is an infinite set.

**Definition 1** (Bayes error rate [50])**.** Consider an $M$-class classification problem, where an input feature $\boldsymbol{x} \in \mathcal{X}$ has to be classified into a class $c \in \mathcal{C} := [M]$. The BER is defined as

$$P_e = \inf_{\phi \in \Phi} \mathcal{E}(\phi) = \inf_{\phi \in \Phi} \mathbb{E}\left[\mathbb{1}\{\phi(\mathbf{X}) \neq C\}\right], \tag{1}$$

where $\Phi$ is the set of all measurable functions $\phi : \mathcal{X} \to \mathcal{C}$, and the expectation is taken over $P_{\mathbf{X},C}$.

The misclassification probability $\mathcal{E}(\phi)$ depends on the quality of the classifier $\phi$ and the BER is obtained by choosing an optimal classifier. In fact, an optimal classifier is theoretically equivalent to the Maximum a Posteriori (MAP) classifier [50], that is, $\phi_{\mathrm{MAP}}(\boldsymbol{x}) = \arg\max_{k \in \mathcal{C}} \Pr(C = k | \mathbf{X} = \boldsymbol{x})$. Plugging the MAP classifier into (1), the BER can be written as

$$P_e = \mathbb{E}\left[1 - \max_{k \in \mathcal{C}} \Pr(C = k | \mathbf{X})\right], \tag{2}$$

where the expectation is taken with respect to $P_{\mathbf{X}}$. Note that $P_e \in \left[0, 1 - \frac{1}{M}\right]$.

Our main objective is to effectively estimate the BER from the dataset $\mathcal{D}$. In the remaining of the paper, we let $\psi : \mathcal{D} \to [0, 1]$ denote the estimator of the BER.

## 3 BER estimation with soft labels

Soft labels have several favorable properties (e.g., they help to prevent an overfitting problem, they lead to a well-structured model, and they improve the prediction performance) that make them widely used in machine learning. For example, they are essential in label smoothing and knowledge distillation, which are widely applied methods to improve model performance [81, 97–101].

There exist several types of soft labels and here we assume that a label is soft if it represents the posterior probability. Specifically, we say that $\mathcal{D} = \{(\boldsymbol{x}_i, \boldsymbol{y}_i)\}_{i=1}^n$ is a dataset with soft labels if $\boldsymbol{x}_i \in \mathcal{X}$ and $\boldsymbol{y}_i \in \mathcal{Y}$ are such that

$$\boldsymbol{y}_i = \begin{bmatrix} \Pr(C = 1 | \mathbf{X} = \boldsymbol{x}_i) \\ \Pr(C = 2 | \mathbf{X} = \boldsymbol{x}_i) \\ \vdots \\ \Pr(C = M | \mathbf{X} = \boldsymbol{x}_i) \end{bmatrix}. \tag{3}$$

This is a standard and widely adopted assumption; it has indeed been argued [17, 35, 42, 61, 69, 101] that using soft labels to approximate posterior probabilities enhances model performance.

**Definition 2** (Base BER estimator)**.** The BER estimator $\psi_{\mathrm{soft}}(\mathcal{D})$ is defined as

$$\psi_{\mathrm{soft}}(\mathcal{D}) = \frac{1}{n} \sum_{(\boldsymbol{x}, \boldsymbol{y}) \in \mathcal{D}} \left(1 - \max_{j \in [M]} y_j\right). \tag{4}$$

We will use the base estimator $\psi_{\mathrm{soft}}$ in (4) as a building block for a robust BER estimation. We note that $\psi_{\mathrm{soft}}$ in (4) with $M = 2$ retrieves the estimator proposed in [46]. The next theorem (proof in Appendix B.1) provides three important properties of $\psi_{\mathrm{soft}}$ in (4).

**Theorem 1.** *Assume that $\mathcal{D}$ contains soft labels as defined in* (3)*. Then, $\psi_{\mathrm{soft}}(\mathcal{D})$ satisfies the following properties:*

1. *(Unbiasedness): $\mathbb{E}[\psi_{\mathrm{soft}}(\mathcal{D})] = P_e$, that is, $\psi_{\mathrm{soft}}(\mathcal{D})$ is an unbiased estimator of the BER;*

2. *(Consistency): For any $\delta \in (0, 1)$, it holds that $|\psi_{\mathrm{soft}} - P_e| < \sqrt{\frac{\left(1 - \frac{1}{M}\right)^2}{2n} \ln \frac{2}{\delta}}$ with probability at least $1 - \delta$, that is, $\psi_{\mathrm{soft}}(\mathcal{D})$ is a consistent estimator of the BER;*

3. *(Asymptotic Normality): $\sqrt{n}(\psi_{\mathrm{soft}} - P_e) \xrightarrow{d} \mathcal{N}(0, \mathit{Var}(Y_{M:M}))$ as $n \to \infty$.*

Theorem 1 shows that the BER can be effectively estimated directly from a dataset that contains soft labels as in (3). Moreover, it highlights that the convergence rate of $\psi_{\mathrm{soft}}(\mathcal{D})$ is $n^{-\frac{1}{2}}$, which is indeed the optimal (parametric) convergence rate.

Another important aspect to assess the performance of $\psi_{\text{soft}}(\mathcal{D})$ is to measure how far it is from the BER $P_e$. Since $\psi_{\text{soft}}(\mathcal{D})$ is unbiased (from Theorem 1) this distance can be measured by computing the variance of $\psi_{\text{soft}}(\mathcal{D})$, which is denoted as $\text{Var}(\psi_{\text{soft}})$ and provided by the next proposition (proof in Appendix B.2).

**Proposition 1.** *It holds that* $\text{Var}(\psi_{\text{soft}}) = \frac{1}{n}\text{Var}(Y_{M:M})$ *and* $\text{Var}(\psi_{\text{soft}}) \leq \frac{\left(1-\frac{1}{M}\right)P_e - P_e^2}{n} \leq \frac{\left(1-\frac{1}{M}\right)^2}{4n}$.

The exact computation of $\text{Var}(\psi_{\text{soft}})$ in Proposition 1 requires the knowledge of the order statistics of $\mathbf{Y}$ and hence, of the label distribution $P_{\mathbf{Y}}$. The upper bounds on $\text{Var}(\psi_{\text{soft}})$ are instead distribution-independent. In particular, both upper bounds show that the rate of convergence of $\text{Var}(\psi_{\text{soft}})$ is $1/n$, which is in line with Theorem 1. Moreover, the first upper bound on $\text{Var}(\psi_{\text{soft}})$ implies that $\text{Var}(\psi_{\text{soft}}) \to 0$ when either $P_e \to 0$ (i.e., also known as realizability assumption [76]) or $P_e \to 1 - \frac{1}{M}$ (i.e., labels and features are independent). The first upper bound on $\text{Var}(\psi_{\text{soft}})$ also allows to find an upper bound on $P_e$ which becomes tight when $P_e \to 1 - 1/M$.

### 3.1 Robustness of $\psi_{\text{soft}}$

We consider robustness to outliers, where an outlier is a data sample that is corrupted by high noise. We use the concept of breakdown point [44, 60] to measure the robustness of $\psi_{\text{soft}}$. The breakdown point captures how robust an estimator is with respect to outliers.[2] In the classical definition of breakdown point, the worst estimator (in terms of robustness) for a dataset $\mathcal{D}$ with outliers is defined as $\psi(\mathcal{D}) = \infty$. However, in our setting $\psi_{\text{soft}}(\mathcal{D}) \leq 1 - \frac{1}{M}$ since $\psi_{\text{soft}}(\mathcal{D})$ is an estimate of $P_e$. Because of this, we next adopt an alternative definition for the breakdown point which is commonly used for a bounded parameter space $\Theta$ [45].

**Definition 3** (Breakdown point). Consider an estimator $\psi : \Omega^n \to \Theta$ of $\theta \in \Theta \subseteq \mathbb{R}^L$. Let $\mathcal{D}^{(\kappa)} = \{D_1, \ldots, D_\kappa\}$, where $D_i \in \Omega$, $\forall i \in [\kappa]$, be a clean dataset without outliers, and let $\widetilde{\mathcal{D}}^{(\tau)} = \{\widetilde{D}_1, \ldots, \widetilde{D}_\tau\}$ be a noisy dataset that is composed of $\tau$ noisy data samples that can be arbitrarily replaced by outliers. Then, the breakdown point of $\psi$ is defined as

$$B(\psi) = \min_{\tau:1\leq\tau\leq\kappa}\left\{\frac{\tau}{\kappa+\tau} : \sup\|\psi(\mathcal{D}^{(\kappa+\tau)}) - \psi(\mathcal{D}^{(\kappa)} \cup \widetilde{\mathcal{D}}^{(\tau)})\| \geq \|\text{rad}(\Theta)\|\right\}, \tag{5}$$

where the $\sup$ is taken over all $\mathcal{D}^{(\kappa+\tau)}, \mathcal{D}^{(\kappa)}$ and $\widetilde{\mathcal{D}}^{(\tau)}$ such that $\mathcal{D}^{(\kappa)} \subset \mathcal{D}^{(\kappa+\tau)}$, and $\text{rad}(\Theta)$ is the vector consisting of the $L$ radii of the largest $L$-dimensional ellipsoid in $\Theta$.[3]

$B(\psi)$ in (5) quantifies the value of the breakdown point for an estimator $\psi : \Omega^{\tau+\kappa} \to \Theta$. In particular, an estimator $\psi$ "breaks down" if $\psi(\mathcal{D}^{(\kappa+\tau)})$ changes of at least $\|\text{rad}(\Theta)\|$ when $\tau$ clean data samples are replaced with $\tau$ outliers. The breakdown point is defined as the minimum ratio between the number of outliers ($\tau$) and the total number of data samples ($\kappa + \tau$), such that $\tau$ outliers are sufficient to break the estimator $\psi(\mathcal{D}^{(\kappa+\tau)})$. From Definition 3, it is clear that the higher the value of the breakdown point, the more robust the estimator is. To measure the breakdown point of $\psi_{\text{soft}}$, we start by noting that $\psi_{\text{soft}}$ is the sample mean of $\{1 - \max_{j\in[M]}(\mathbf{y}_i)_j\}_{i=1}^n$ (see (4)). Thus, using the notation as in Definition 3, we have that $\Theta = [0, 1 - \frac{1}{M}]$, which leads to $\text{rad}(\Theta) = \frac{1}{2} - \frac{1}{2M}$. This implies that $B(\psi_{\text{soft}}) = \frac{1}{n}$ (e.g., when $\mathcal{D}$ contains an outlier $\mathbf{y}$ such that $\max_{j\in[M]} y_j = \infty$) and hence, $\psi_{\text{soft}}$ is not robust to outliers.

We next propose a methodology, inspired by the median-of-means estimator [65], to make any BER estimator (and hence, also $\psi_{\text{soft}}$) robust.

**Definition 4** (Median-of-BERs (MoB) estimator). Consider any BER estimator $\psi$ and a dataset $\mathcal{D}$. First, partition $\mathcal{D}$ into $K$ sub-datasets $\mathcal{D}_k$, $k \in [K]$ such that $|\mathcal{D}| = n = cK$, where $c \in \mathbb{N}$ is the number of data samples in $\mathcal{D}_k$. Then, the Median-of-BERs (MoB) estimator is defined as

$$\text{MoB}_K(\psi, \mathcal{D}) = \text{med}(\{\psi(\mathcal{D}_k) : k \in [K]\}), \tag{6}$$

where $\text{med}(\mathcal{S})$ is the sample median of $\mathcal{S}$.

---

[2] The sample mean can fail to estimate the true mean by a single outlier (e.g., a sample $x$ with a value very different from the true mean). Thus, the sample mean has a breakdown point equal to $1/n$, meaning that one outlier can break the estimate. Differently, the median is more robust since it has a breakdown point of $1/2$, i.e., half of the samples need to be outliers for breaking the estimate.

[3] Any $L$-dimensional ellipsoid (after rotation) can be defined as $\frac{x_1^2}{a_1^2} + \frac{x_2^2}{a_2^2} + \ldots + \frac{x_L^2}{a_L^2} = 1$. Then, the $L$ radii are $a_1, a_2, \ldots, a_L$.

The next theorem (proof in Appendix B.3) provides three important properties of $\mathsf{MoB}_K(\psi_{\mathrm{soft}}, \mathcal{D})$.

**Theorem 2.** *Assume that $\mathcal{D}$ contains soft labels as defined in* (3). *Then, the MoB estimator* $\mathsf{MoB}_K(\psi_{\mathrm{soft}}, \mathcal{D})$ *satisfies the following properties:*

1. (Consistency): *For all $t \lesssim K$, with probability at least $1 - 4e^{-2t}$, it holds that*

$$\left|\mathsf{MoB}_K(\psi_{\mathrm{soft}}, \mathcal{D}) - P_e\right| \leq \left(\frac{\left(1 - \frac{1}{M}\right)^3}{2\sqrt{3}\mathit{Var}(Y_{M:M})} \frac{K}{\sqrt{n-K}} + 3\sqrt{t\mathit{Var}(Y_{M:M})}\right)\sqrt{\frac{1}{n-K}}, \quad (7)$$

   *and hence, $\mathsf{MoB}_K(\psi_{\mathrm{soft}}, \mathcal{D})$ is a consistent estimator of the BER;*

2. (Breakdown): *Its breakdown point is given by $B\left(\mathsf{MoB}_K(\psi_{\mathrm{soft}}, \mathcal{D})\right) = \left\lfloor \frac{K+1}{2} \right\rfloor \frac{1}{n}$;*

3. (Asymptotic Normality): *If $K \to \infty$ and $K = o(\sqrt{n})$ as $n \to \infty$, then* $\sqrt{n}\left(\mathsf{MoB}_K(\psi_{\mathrm{soft}}, \mathcal{D}) - P_e\right) \xrightarrow{d} \mathcal{N}\left(0, \frac{\pi}{2}\mathit{Var}(Y_{M:M})\right)$.

Theorem 2 shows that $\mathsf{MoB}_K(\psi_{\mathrm{soft}}, \mathcal{D})$ has the same rate of convergence of $n^{-\frac{1}{2}}$ as $\psi_{\mathrm{soft}}$ (see Theorem 1). However, $\mathsf{MoB}_K(\psi_{\mathrm{soft}}, \mathcal{D})$ has a higher breakdown point (and hence, is more robust) than $\psi_{\mathrm{soft}}$. Nonetheless, this robustness is attained at two expenses: (i) $\mathsf{MoB}_K(\psi_{\mathrm{soft}}, \mathcal{D})$ is not an unbiased estimator of $P_e$; and (ii) the variance of $\mathsf{MoB}_K(\psi_{\mathrm{soft}}, \mathcal{D})$ is larger than the one of $\psi_{\mathrm{soft}}$ by a factor of $\pi/2$ in the asymptotic regime.

Theorem 2 also points out to an interesting trade-off, with respect to $K$, between accuracy and robustness. For example, setting $K = \sqrt{n}$ leads to a higher breakdown point (and hence, is more robust) than setting $K = \ln n$. However, the concentration bound in (7) (which measures the accuracy of the estimation) has a much larger value with $K = \sqrt{n}$ than with $K = \ln n$.

## 4 BER estimation with noisy labels

The soft label data discussed in Section 3 might be challenging to obtain as data labels are often corrupted by noise or other perturbations. In such scenarios, data labels are unreliable [27, 64, 68] and several studies have been conducted on them [37, 57, 80, 88, 90, 92, 94, 96].

With the goal to broaden the applicability of $\psi_{\mathrm{soft}}$ (or its robust version $\mathsf{MoB}_K(\psi_{\mathrm{soft}}, \mathcal{D})$), we here study their performance in scenarios where the soft labels are corrupted by various typical types of noise.[4] We denote by $\widetilde{\mathbf{Y}} \sim P_{\widetilde{\mathbf{Y}}|\mathbf{Y}}$ the noisy soft label, i.e., a noisy soft label $\widetilde{\mathbf{Y}} = \tilde{\mathbf{y}}_i$ is distributed according to the conditional probability distribution $P_{\widetilde{\mathbf{Y}}|\mathbf{Y}=\mathbf{y}_i}$, where $\mathbf{y}_i$ is the true soft label.

### 4.1 Additive noise on the data labels

We here consider the case where the labels are corrupted by additive noise, i.e., we have $(\boldsymbol{x}, \tilde{\boldsymbol{y}}) \in \widetilde{\mathcal{D}}_{\mathrm{A}}$ where $\tilde{\boldsymbol{y}} = \boldsymbol{y} + \boldsymbol{z}$, where $\boldsymbol{z}$ is an $M$-dimensional random noise vector. We assume that $\boldsymbol{z}$ has i.i.d. components each with zero mean.

We start our analysis by showing that $\psi_{\mathrm{soft}}$ in (4) applied on $\widetilde{\mathcal{D}}_{\mathrm{A}}$ is neither a consistent nor an unbiased estimator of $P_e$. We, in fact, have that

$$\psi_{\mathrm{soft}}(\widetilde{\mathcal{D}}_{\mathrm{A}}) = 1 - \frac{1}{n}\sum_{i=1}^{n}\max_{j\in[M]}\{(\boldsymbol{y}_i)_j + (\boldsymbol{z}_i)_j\}, \quad (8)$$

which, by the law of large numbers, converges to

$$\lim_{n\to\infty}\psi_{\mathrm{soft}}(\widetilde{\mathcal{D}}_{\mathrm{A}}) = 1 - \mathbb{E}\left[\max_{j\in[M]}\{Y_j + Z_j\}\right]. \quad (9)$$

Then, the inequalities $\max_i x_i + \min_j y_j \leq \max_i\{x_i + y_i\} \leq \max_i x_i + \max_j y_j$ imply that

$$\mathbb{E}[Z_{1:M}] \leq P_e - \lim_{n\to\infty}\psi_{\mathrm{soft}}(\widetilde{\mathcal{D}}_{\mathrm{A}}) \leq \mathbb{E}[Z_{M:M}], \quad (10)$$

---

[4]We refer an interested reader to Appendix B.4 where a noise that randomly shuffles the labels is also studied.

which follow from (2) and (3). The bounds in (10) demonstrate that with labels corrupted by additive noise, $\psi_{\text{soft}}$ is (in general) an inconsistent estimator of $P_e$. By following similar steps, it can be easily proved that $\psi_{\text{soft}}$ is also a biased estimator of $P_e$. In particular, the bounds in (10) show that $\psi_{\text{soft}}$ has a bias which is bounded by the expected value of the smallest (lower bound) and of the largest (upper bound) order statistics of the noise. In what follows, we provide two examples of distributions for which bounds on these expected values have been computed.

*Example* 1. Let $\mathbf{Z} \overset{i.i.d.}{\sim} \text{Uniform}(-a, a)$. Then, $\mathbb{E}[Z_{i:M}] = \frac{2ai}{M+1} - a$, which as $n \to \infty$ implies that $|P_e - \psi_{\text{soft}}(\widetilde{\mathcal{D}}_A)| \leq a\frac{M-1}{M+1}$.

*Example* 2. Let $\mathbf{Z} \overset{i.i.d.}{\sim} \gamma^2$-sub-Gaussian.[5] Then, we have that $\mathbb{E}[Z_{M:M}] = -\mathbb{E}[Z_{1:M}]$ and $\mathbb{E}[Z_{M:M}] \leq \sqrt{2\gamma^2 \ln M}$ [9] which as $n \to \infty$ implies that $|P_e - \psi_{\text{soft}}(\widetilde{\mathcal{D}}_A)| \leq \sqrt{2\gamma^2 \ln M}$.

Our analysis above shows that $\psi_{\text{soft}}(\widetilde{\mathcal{D}}_A)$ is (in general) an inconsistent and biased estimator of $P_e$. Because of this, we next propose a new BER estimator, which (different from $\psi_{\text{soft}}$) leverages the features $\{\boldsymbol{x}_i\}_{i=1}^n$ to denoise $\tilde{\boldsymbol{y}}_i$. We refer to this estimator as $\psi_{\text{DN}}$. Our main intuition behind proposing $\psi_{\text{DN}}$ stems from the fact that data samples having the same feature $\boldsymbol{x}$ should be labeled with the same (or at least similar) label. This can be attained by noting that, even if labels are noisy, it is possible to minimize the effect of a zero-mean noise by averaging the noisy labels associated with the same feature. We next formally define our denoising BER estimator $\psi_{\text{DN}} : \mathbb{R}^M \to [0, 1]$.

**Definition 5** (Denoise estimator). For a noisy dataset $\widetilde{\mathcal{D}}_A = \{(\boldsymbol{x}_i, \tilde{\boldsymbol{y}}_i)\}_{i=1}^n$, let the denoised label $\boldsymbol{s}(\boldsymbol{x}_i)$ for all $i \in [n]$, be defined as

$$\boldsymbol{s}(\boldsymbol{x}_i) = \frac{\sum_{j=1}^n \mathbb{1}\{\boldsymbol{x}_j = \boldsymbol{x}_i\}\tilde{\boldsymbol{y}}_j}{\sum_{j=1}^n \mathbb{1}\{\boldsymbol{x}_j = \boldsymbol{x}_i\}}, \tag{11}$$

and let $\text{idx}(\boldsymbol{x}_i) = \arg\max_{j \in [M]}\{(\boldsymbol{s}(\boldsymbol{x}_i))_j\}$. Then, the denoise BER estimator is defined as

$$\psi_{\text{DN}}(\widetilde{\mathcal{D}}_A) = \frac{1}{n}\sum_{i=1}^n \left(1 - (\tilde{\boldsymbol{y}}_i)_{\text{idx}(\boldsymbol{x}_i)}\right). \tag{12}$$

The next theorem (proof in Appendix B.5) provides some important properties of $\psi_{\text{DN}}(\widetilde{\mathcal{D}}_A)$.

**Theorem 3.** *Let $\widetilde{\mathcal{D}}_A = \{(\boldsymbol{x}_i, \tilde{\boldsymbol{y}}_i)\}_{i=1}^n$ be a dataset that consists of noisy soft labels $\tilde{\boldsymbol{y}}_i = \boldsymbol{y}_i + \boldsymbol{z}_i$ where $\boldsymbol{z}_i \overset{i.i.d.}{\sim} P_{\mathbf{Z}}$ is the zero mean noise. Then, $\psi_{\text{DN}}(\widetilde{\mathcal{D}}_A)$ is a consistent estimator of $P_e$.*

*Moreover, if the noise has bounded support, that is $\Pr(\mathbf{Z} \in [a, b]^M) = 1$, then*

1. *(Asymptotical unbiasedness): $\psi_{\text{DN}}(\widetilde{\mathcal{D}}_A)$ is an asymptotically unbiased estimator of $P_e$;*

2. *(Denoising Consistency): For any $\delta \in (0, 1)$, it holds that $|\boldsymbol{s}(\boldsymbol{x}) - \boldsymbol{y}| < \sqrt{\frac{\left(1 - \frac{1}{M} - a + b\right)^2}{2n_{\boldsymbol{x}}} \ln \frac{2}{\delta}}$ with probability at least $1 - \delta$, where $n_{\boldsymbol{x}} = \sum_{j=1}^n \mathbb{1}\{\boldsymbol{x}_j = \boldsymbol{x}\}$ and $\boldsymbol{s}(\boldsymbol{x})$ is the denoised label defined in (11).*

The results in Theorem 3 broadens the applicability of an effective BER estimator to a larger class of datasets, e.g., to scenarios where the dataset includes multiple data samples representing the same feature. For example, $\psi_{\text{DN}}$ works well on a noisy dataset where each data sample has multiple labels. Moreover, as we will see in Section 4.2, $\psi_{\text{DN}}$ is also useful for one-hot labels. We also expect that $\psi_{\text{DN}}$ will work well with privatized datasets where noise was added to enhance privacy [26].

*Remark* 1. Our estimator $\psi_{\text{DN}}$ only requires the dataset $\widetilde{\mathcal{D}}_A$. This is a more practical approach than the one in [46] which instead also requires the true one-hot label. The estimator in [46], which is referred to as a genie estimator $\psi_{\text{genie}}$, can be obtained by replacing $\boldsymbol{s}(\boldsymbol{x})$ in (11) with the true one-hot label. In Appendix B.6, we provide properties of $\psi_{\text{genie}}(\widetilde{\mathcal{D}}_A)$ that generalize the results in [46] for any $M \geq 2$. As expected, $\psi_{\text{genie}}$ is a consistent and unbiased estimator of $P_e$.

---

[5]We say that a random variable $Z$ is $\gamma^2$-sub-Gaussian if $\mathbb{E}[e^{\lambda(Z - \mathbb{E}[Z])}] \leq e^{\frac{\lambda^2\gamma^2}{2}}$, $\forall\lambda \in \mathbb{R}$.

## 4.2 One-hot labels

A dataset consisting of one-hot labels, denoted as $\overline{\mathcal{D}} = \{(\boldsymbol{x}_i, \overline{\boldsymbol{y}}_i)\}_{i=1}^n$, contains little information about the class posterior probability. We assume that the one-hot label is constructed from the corresponding soft label as follows,

$$\overline{\mathbf{Y}} = \mathbf{e}_i \text{ with probability } Y_i, \tag{13}$$

where: (i) $\overline{\mathbf{Y}}$ is the one-hot random vector; (ii) $\mathbf{Y}$ is the soft label random vector in (3); and (iii) $\mathbf{e}_i \in \mathbb{R}^M$ is the standard basis vector with a one in the $i$-th position and zero in all the other entries.

We can think of the one-hot label construction in (13) as a noisy label. Specifically, the noise $\mathbf{Z} \in \{\mathbf{e}_i - \boldsymbol{y}\}_{i=1}^M$ with the probability mass function $p_{\mathbf{Z}|\mathbf{Y}}(\boldsymbol{z}|\boldsymbol{y}) = \sum_{j=1}^M y_j \mathbb{1}\{\boldsymbol{z} = \mathbf{e}_j - \boldsymbol{y}\}$ satisfies $\overline{\mathbf{Y}} = \mathbf{Y} + \mathbf{Z}$. It is not difficult to see that $\mathbb{E}[\mathbf{Z}] = \mathbb{E}[\mathbb{E}[\mathbf{Z}|\mathbf{Y}]] = \mathbf{0}$ and hence, our denoise estimator $\psi_{\mathrm{DN}}(\widetilde{\mathcal{D}}_A)$ in (12), or its robust version $\mathsf{MoB}_K(\psi_{\mathrm{DN}}, \widetilde{\mathcal{D}}_A)$, can estimate the BER also for the case of one-hot labels. However, we also note that with the above construction, the labels might be too noisy to estimate the BER in practice. This suggests that a more refined denoising method than the one in Definition 5 for one-hot labels might be needed for a more effective BER estimation.

Motivated by the above discussion, we next propose a denoising method for one-hot labels, which leverages neighbor samples to mitigate the noise.[6] Our choice of such a denoising method mainly stems from the fact that one would expect that neighboring samples should have similar posterior probabilities. In particular, for each $\boldsymbol{x}_i, i \in [n]$, we consider all of its neighbors (features that are at most at a distance $r$ from $\boldsymbol{x}_i$) and we average the corresponding noisy labels in a spirit similar to (11). We next formally define our denoising BER estimator, which we refer to as $\psi_{\mathrm{C}}$.

**Definition 6** (Cluster denoise estimator). Given $\mathsf{d} : \mathcal{X}^2 \to \mathbb{R}_+$ and $r \in \mathbb{R}_+$, define $\mathsf{Cluster}_i := \{(\boldsymbol{x}, \boldsymbol{y}) \in \widetilde{\mathcal{D}}_A : \mathsf{d}(\boldsymbol{x}_i, \boldsymbol{x}) \le r\}$. Then, the denoised label for $\tilde{\boldsymbol{y}}_i$, for all $i \in [n]$, is defined as

$$\hat{\boldsymbol{y}}_i = \frac{\sum_{(\boldsymbol{x}, \tilde{\boldsymbol{y}}) \in \mathsf{Cluster}_i} \tilde{\boldsymbol{y}}}{|\mathsf{Cluster}_i|}, \tag{14}$$

and the cluster-based BER estimator is defined as

$$\psi_{\mathsf{C}}(\widetilde{\mathcal{D}}_A, \mathsf{d}, r) = \frac{1}{n} \sum_{i=1}^n \left(1 - \max_{j \in [M]} \{(\hat{\boldsymbol{y}}_i)_j\}\right). \tag{15}$$

*Remark* 2. Preprocessing the dataset that aggregates data samples of similar features can also be helpful in improving the BER estimate for noisy labels, and it can be paired with the cluster denoise estimator in Definition 6. A naive approach would consist of reducing the feature dimensionality and increasing the number of data samples inside clusters (14), which helps mitigate the effect of the noise. Some notable examples of data preprocessing are the classical principal component analysis (PCA) [77] and the representation learning [5] (e.g., variational auto encoder [52]).

*Remark* 3. Another viewpoint to the denoising method in (14) is the connection to the non-parametric estimation of the conditional expectation $\mathbb{E}[\mathbf{Y}|\mathbf{X} = \boldsymbol{x}]$, where $\mathbf{Y}$ is the true soft label corresponding to $\mathbf{X} = \boldsymbol{x}$. In particular, estimating $\mathbb{E}[\mathbf{Y}|\mathbf{X} = \boldsymbol{x}]$ is equivalent to denoising the noisy label associated with $\boldsymbol{x}$ as in (14). Under such a viewpoint, the BER estimator in Definition 6 resembles the Nadaraya–Watson estimator with Parzen window kernel [63, 89].

## 5 Experiments

We empirically validate our results using various datasets, namely: 1) a synthetic dataset with different noises, including one-hot labels; 2) two benchmark datasets CIFAR-10H [4] and Fashion-MNIST-H [46]; and 3) MovieLens [38], a real-world dataset for movie recommendations. We compare our estimators with two SOTA BER bounds [72], namely the generalized Henze-Penrose (GHP) BER bounds [75] and the $k$-Nearest Neighbor (NN) BER bounds [16]. Details on the $k$-NN bounds and GHP bounds, and additional results can be found in Appendix C.

---

[6]Several works have taken advantage of neighbor samples to estimate the BER or the probability divergence. We refer an interested reader to [67] and references therein.

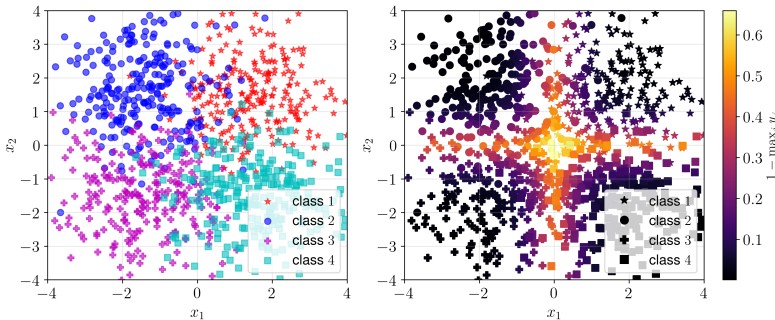

Figure 1: 4-classes Gaussian data samples. The left figure shows the samples of each class, and the right figure displays the corresponding soft label information as the maximum value of the soft label.

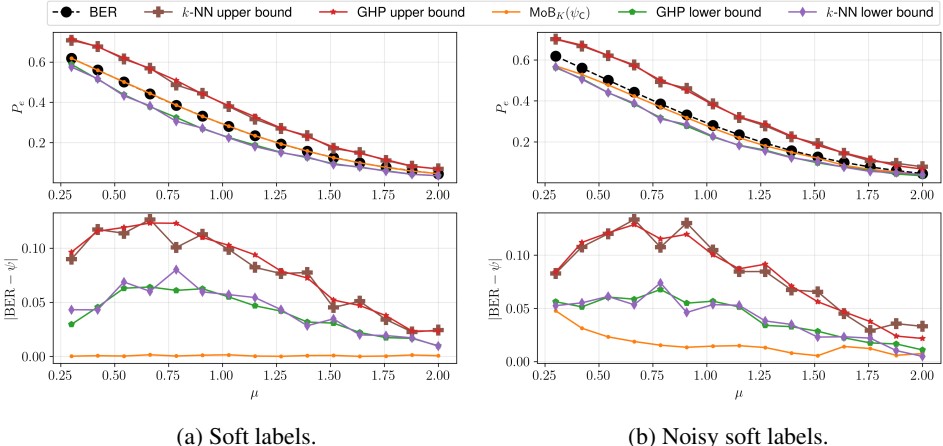

(a) Soft labels.          (b) Noisy soft labels.

Figure 2: Comparison of $\mathsf{MoB}_K(\psi_{\mathsf{C}})$ where $\psi_{\mathsf{C}}$ is defined in (15) with the GHP bounds and the $k$-NN bounds with $k = 1$. We consider $n = 2,000$ samples generated as in Figure 1 with noise $\mathbf{Z} \sim \mathcal{N}\left(\mathbf{0}, \frac{1}{5}I_4\right)$. $\mathsf{MoB}_K(\psi_{\mathsf{C}})$ uses $K = \lfloor \sqrt{n} \rfloor$ (with this choice, Theorem 2 ensures the asymptotic normality of $\mathsf{MoB}_K$), the Euclidean distance for d, and $r = 1/5$.

**Synthetic dataset with soft labels and noisy labels.** We consider a 4-class classification problem with equiprobable classes $C \in \mathcal{C}_\mu := \{(\mu, \mu), (-\mu, \mu), (-\mu, -\mu), (\mu, -\mu)\}$, where $\mu > 0$ is a parameter that controls the classification hardness. We generate the feature $\mathbf{X} \in \mathbb{R}^2$ according to a 2-dimensional Gaussian distribution with mean $\boldsymbol{c}$ (i.e., a realization of $C \in \mathcal{C}_\mu$) and covariance matrix $I_2$. The corresponding soft labels $\boldsymbol{y}_i$'s are then obtained by the Bayes' theorem. In particular, since $f_{\mathbf{X}|C} \sim \mathcal{N}(C, I_2)$ and $P_C(\boldsymbol{c}) = 1/4$ if $\boldsymbol{c} \in \mathcal{C}_\mu$ (and zero otherwise), according to (3) we have that $(\boldsymbol{y}_i)_k = \frac{f_{\mathbf{X}|C}(\boldsymbol{x}_i|\boldsymbol{c})P_C(\boldsymbol{c})}{\sum_{\alpha \in \mathcal{C}_\mu} f_{\mathbf{X}|C}(\boldsymbol{x}_i|\alpha)P_C(\alpha)}$.[7] Figure 1 illustrates $n = 1,000$ features (left figure), and the corresponding soft labels (right figure) for the synthetic dataset generated as explained above; in particular, for each feature $\boldsymbol{x}_i$, we only reported the label $(\boldsymbol{y}_i)_{j^\star}$ where $j^\star = \arg\max_{j \in [4]} (\boldsymbol{y}_i)_j$ which is required for evaluating $\psi_{\text{soft}}$ in (4). As expected, a point near the decision boundary (lines for $x_1 = 0$ or $x_2 = 0$) has a smaller maximum value in its soft label.

The results in Figure 2 empirically demonstrate the effectiveness of our estimators, which consistently outperform the others.[8] Moreover, our estimators estimate the exact value of the BER, while the others derive upper and lower bounds on the BER, which might not be tight.

---

[7]Without loss of generality, we consider the following mapping: $k = 1$ if $\boldsymbol{c} = (\mu, \mu)$; $k = 2$ if $\boldsymbol{c} = (-\mu, \mu)$; $k = 3$ if $\boldsymbol{c} = (-\mu, -\mu)$; and $k = 4$ if $\boldsymbol{c} = (\mu, -\mu)$.

[8]Among the proposed estimators, in Figure 2 we only evaluated $\mathsf{MoB}_K(\psi_{\mathsf{C}})$ since we observed that it performs well with respect to SOTA methods and hence, we omitted the other estimators.

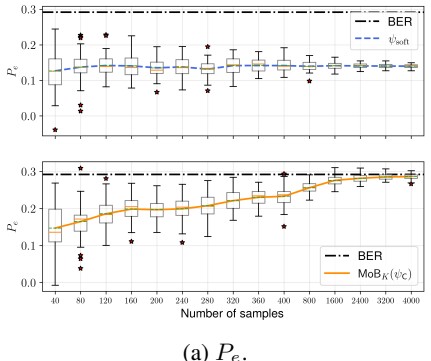
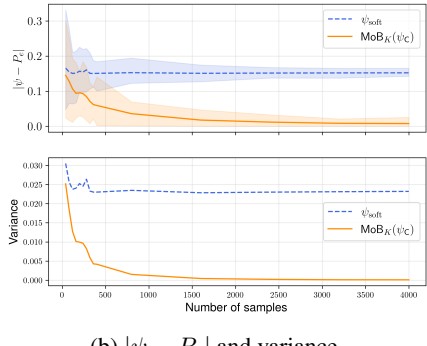

(a) $P_e$.

(b) $|\psi - P_e|$ and variance.

Figure 3: Comparison of $\mathsf{MoB}_K(\psi_\mathsf{C})$ (evaluated with $\psi_\mathsf{C}$ in (15)) with $\psi_{\mathrm{soft}}$ on noisy soft labels. We consider $\mathbf{Z} \sim \mathcal{N}\left(\mathbf{0}, 1/5 I_4\right)$. To model outliers, we added $\tilde{Z} = 0.2 \cdot \mathrm{Ber}\left(1/2\right)$ with probability $1/10$ to each entry of the soft labels. For different $n$, the parameters of $\mathsf{MoB}_K(\psi_\mathsf{C})$ are chosen as $K = \lfloor\sqrt{n}\rfloor$, d is the Euclidean distance, and $r = 1/5$. We iterate the experiment 50 times for each $n$.

We further conducted experiments to verify the effectiveness of our denoising and robustifying methods on data samples with label noise and outliers. Figure 3 shows the comparison of the denoising estimator $\mathsf{MoB}_K(\psi_\mathsf{C})$ with the base estimator $\psi_{\mathrm{soft}}$. We observe that $\psi_{\mathrm{soft}}$ suffers from the label noise, which leads to a bias in the estimation. However, our denoising estimator $\mathsf{MoB}_K(\psi_\mathsf{C})$ suitably denoises the noisy labels and effectively estimates the BER. As shown in Figure 3b, in fact, the absolute error between the estimate and the BER as well as the variance decrease as $n$ grows.

**Synthetic dataset with one-hot labels.** We analyze the performance of $\mathsf{MoB}_K(\psi_\mathsf{C})$ in the same Gaussian setting described above, but with one-hot labels. Each soft label of the samples is mapped to a one-hot label according to the categorical distribution with probabilities equal to the soft labels (see (13)). Figure 4 shows that our denoising method is capable of suitably reducing the noise in one-hot labels; in fact, it correctly estimates the BER after around $10^4$ samples when $\mathsf{MoB}_K(\psi_\mathsf{C})$ converges to the BER. From Figure 4, we also observe that $\psi_{\mathrm{soft}}$ performs poorly; this suggests that denoising methods are essential for the BER estimation task, hence worth further investigation.

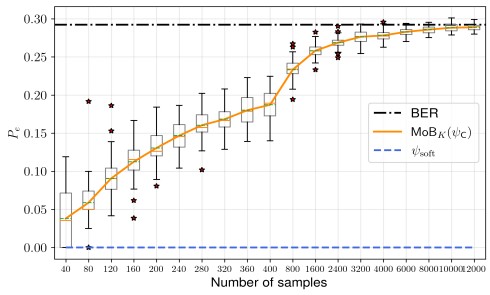

Figure 4: Comparison of $\mathsf{MoB}_K(\psi_\mathsf{C})$ with $\psi_{\mathrm{soft}}$ for Gaussian samples with one-hot labels.

**CIFAR-10H [4] and Fashion-MNIST-H [46].**
The CIFAR-10H dataset is a variation of CIFAR-10 [53], constructed by labeling $10^4$ images in the test dataset of CIFAR-10 by multiple labelers. The Fashion-MNIST-H dataset is populated by $10^4$ images in the test dataset of Fashion-MNIST [93] in a similar manner. These two datasets have 10 classes, but we further categorized these into $M \in \{2, 3\}$ classes. In particular, for CIFAR-10H, we assigned $C_1 = \{\mathrm{airplane, automobile, ship, truck}\}$, $C_2 = \{\mathrm{bird, cat, deer, dog, frog, horse}\}$ when $M = 2$, and $C_1 = \{\mathrm{airplane, automobile, ship, truck}\}$, $C_2 = \{\mathrm{cat, dog, frog}\}$, $C_3 = \{\mathrm{bird, deer, horse}\}$ when $M = 3$. Similarly, for Fashion-MNIST-H, we assigned $C_1 = \{\mathrm{t\text{-}shirt/top, pullover, dress, coat, shirt}\}$, $C_2 = \{\mathrm{trouser, sandal, sneaker, bag, ankleboot}\}$ when $M = 2$, and $C_1 = \{\mathrm{t\text{-}shirt/top, pullover, coat, shirt}\}$, $C_2 = \{\mathrm{trouser, dress, bag}\}$, and $C_3 = \{\mathrm{sandal, sneaker, ankleboot}\}$ when $M = 3$.

From Table 1, we observe that the estimates using $\mathsf{MoB}_K(\psi_{\mathsf{DN}})$ with $K = \lfloor\sqrt{n}\rfloor$ are slightly lower than $\psi_{\mathsf{DN}}$, which is due to the robustness of $\mathsf{MoB}_K(\psi_{\mathsf{DN}})$ to outliers.[9] We also highlight that the BER estimates can be leveraged to determine which task is more difficult. Intuitively, performing

---

[9] In general, there might be outliers that lead to low values of the estimated BER. However, since the labels belong to $[0, 1]$ and the estimated BER values in our examples are very small, it is reasonable to assume that outliers with large values would be more impactful than outliers with small values.

Table 1: BER estimate on benchmark and real-world datasets.

| # classes | CIFAR-10H | | | Fashion-MNIST-H | | | MovieLens | |
|---|---|---|---|---|---|---|---|---|
| | 2 | 3 | 10 | 2 | 3 | 10 | 2 | 3 |
| $\psi_{\mathsf{DN}}$ | 0.0050 | 0.0177 | 0.0456 | 0.0348 | 0.0932 | 0.2825 | 0.3063 | 0.4031 |
| $\mathsf{MoB}_K(\psi_{\mathsf{DN}})$ | 0.0044 | 0.0168 | 0.0440 | 0.0347 | 0.0931 | 0.2816 | 0.3065 | 0.4035 |

the classification task over CIFAR-10H is much easier than performing it over Fashion-MNIST-H, and this is supported by the results in Table 1, i.e., the BERs associated with CIFAR-10H are smaller than those over Fashion-MNIST-H. It is worth noting that test dataset overfitting can happen when benchmark datasets are considered [3, 46, 51, 59, 91]. This can also be observed by the results in Table 1, where our BER estimates of the 10-class classifications are larger than the SOTA error rates of 0.005 on CIFAR-10 by [24] and of 0.0309 on Fashion-MNIST by [82]. We also suspect that, since the labels are assigned by humans, who might not be experts, the datasets might still have a considerable amount of label noise, which would lead to incorrect BER estimates. However, no estimators (including ours) can estimate the BER from a dataset that contains very noisy labels.

**MovieLens [38].** This dataset consists of 25 million ratings to $62,000$ movies by $162,000$ users. Each rating ranges from $0.5$ to $5$ with step size $0.5$. We first considered a movie classification task: a user either likes a movie or not. This is a complex binary classification task with the input feature being a movie (in general, a two-hour long video with audio) and the class being either $0$ (dislike the movie) or $1$ (like the movie). To ensure that a label belongs to $[0, 1]$, we applied min-max normalization to each rating. Moreover, in order to have enough ratings for each movie when we applied the denoising method, we filtered out some data if the number of ratings was smaller than $100$. Then, $\psi_{\mathsf{DN}}$ and $\mathsf{MoB}_K(\psi_{\mathsf{DN}})$ estimated the BER of such movie classification task, and they yielded a BER of around $0.3$. We then categorized the ratings into $M = 3$ classes, i.e., we assigned $C_1 = \{0.5, 1.0, 1.5\}$, $C_2 = \{2.0, \ldots, 3.5\}$ and $C_3 = \{4.0, 4.5, 5.0\}$. On this task, the BER estimate is around $0.4$. These BERs are quite large, implying that the movie classification is a hard task to perform. This may be justified by the fact that ratings of movies are subjective, and movie recommendations are indeed challenging without users' information (e.g., content-based [47] or item-based [74] recommendation systems use users' information).

## 6    Conclusion

In this paper, we investigated the challenge of estimating the BER for multi-class classifications. We proposed a BER estimator, $\psi_{\mathsf{soft}}$, which we proved to be unbiased, consistent, and asymptotically normal when applied to soft-labeled datasets. By leveraging the median-of-mean method, we also proposed a methodology to make any BER estimate robust. To ensure the applicability of the BER estimator in practical scenarios, we analyzed the challenges posed by noisy soft labels, including those with additive noise and one-hot labels. For such noisy labeled datasets, we developed denoising techniques that effectively mitigate the label noise by using the corresponding features. We showed that these denoising BER estimators are unbiased and consistent under mild noise assumptions. Our experimental results, drawn from synthetic and real-world datasets, validated our theoretical results.

Although in this work we assumed that the set of features is finite, several of our results (e.g., Theorem 1 and Theorem 2) can be easily shown to hold also for the infinite feature space. A research direction worth further investigation would consist of proving that the cluster denoise estimator in Definition 6 has similar appealing properties (see Theorem 3) as the denoise estimator in Definition 5 (which assumes that the set of the features is discrete). Always along these lines, a second interesting future work would be to analyze the rate of convergence of our BER estimator paired with the denoising method as a function of the characteristics (e.g., cardinality, distribution) of the feature space. Such an analysis would indeed provide insights into the difficulty of a classification task. Another relevant avenue for future research lies in assessing the optimal performance of multi-label learning, where a single data point can be associated with multiple classes. While adapting our framework to predict the minimum Hamming loss for multi-label learning might be relatively straightforward, estimating other metrics (e.g., the F1 score and the exact match ratio) might be a difficult task, potentially necessitating new methodologies.

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

# Appendices

## A   Related work

It is well known that the BER of a binary classification problem can be achieved by a likelihood ratio test [66] and the BER of a multiclass classification problem can be attained by a Maximum a Posteriori (MAP) classifier [50]. The problem of estimating the BER using a dataset has been extensively investigated from the middle of the 20th century [16] to the present [13], leveraging different approaches, such as the $k$-nearest neighbors (NN) method, probabilistic divergence, and bounds on the BER. In the early stage, the main approach was to derive upper and lower bounds on the BER, while recent work mainly focuses on directly estimating the BER rather than deriving bounds on it.

**BER bounds.**   Several bounds on the BER in binary classification tasks have been proposed using probability divergence and distance metrics. Notable approaches use the Mahalanobis distance [20, 86], the Bhattacharyya distance between two class conditional distributions [20, 49, 86], the Chernoff coefficient (a generalization of the Bhattacharyya distance) [14, 25, 29], the Henze-Penrose divergence [6], the Jensen-Shannon divergence [55], bounds for the $\min\{a, b\}$ function [2, 39] (a generalization of the equivocation bound [41] and of the Bayesian bound [12]), the $M_0$-distance [85], the Fano's inequality [15], and the $L_2$ norm of posterior probabilities [19]. Some of these bounds are tight, but they still require estimating the data distribution (e.g., class conditional distributions or posterior probabilities), which may not be practical, particularly for modern data (e.g., images and videos).[10] Bounds on the BER were also derived using non-parametric approaches [16, 21, 30, 33], such as the $k$-NN classifier. A remarkable advantage of a $k$-NN classifier for this problem is its distribution-free property to bound the BER (i.e., it works under no assumption on the data distribution). A convergence rate analysis for universal BER estimators was developed in [1, 22].

**Direct estimation of the BER.**   As an alternative approach to using bounds on the BER, some recent research directly estimates the BER [13, 46, 67, 83]. In particular, the authors in [67] expressed the BER as a function of the $f$-divergence and proposed a density ratio estimator to evaluate such an $f$-divergence that yields the BER. Under various label assumptions, a direct BER estimator was proposed in [46]; this estimator is unbiased and consistent for soft labels, and it has desirable properties. Another direct estimator for the BER was proposed in [13]. In particular, this estimator uses the relationship between the BER and the miss-classified samples.

**Extension to multi-class classification.**   The extension of the BER in binary classification to multi-class classification has been extensively investigated both in terms of generalizing existing bounds and deriving new approaches. For example, the Chernoff bound [34], the Jensen-Shannon divergence [55], the Fano's inequality [15], the general mean distance [8], the $L_2$ norm of posterior probabilities [19], and the generalized Henze-Penrose divergence [75] can be used to derive bounds on the BER in multi-class classification problems. Non-parametric methods [31, 58], such as the $k$-NN classifier, can also bound the BER in multi-class classification settings. Bounds based on the error probability of a $k$-NN classifier in the asymptotic regime were studied in [16], and some bounds in the non-asymptotic regime can be found in [10, 32].

---

[10]Interested readers are referred to [2, 11, 56, 72]. In particular, comparisons of a variety of upper bounds on the BER were studied in [56, 72], and an arbitrary tight upper bound was proposed in [2].

## B  Proofs and auxiliary results

### B.1  Proof of Theorem 1

*Proof.* By taking the expectation of $\psi_{\text{soft}}$ in (4) with respect to $P_{\mathbf{X},\mathbf{Y}}$, we obtain

$$
\begin{aligned}
\mathbb{E}[\psi_{\text{soft}}(\mathcal{D})] &= \mathbb{E}\left[\frac{1}{n}\sum_{(\boldsymbol{x},\boldsymbol{y})\in\mathcal{D}}\left(1 - \max_{j\in[M]} y_j\right)\right] \\
&= \frac{1}{n}\sum_{(\boldsymbol{x},\boldsymbol{y})\in\mathcal{D}}\mathbb{E}\left[1 - \max_{j\in[M]}\Pr(C=j|\mathbf{X})\right] \\
&= 1 - \mathbb{E}\left[\max_{j\in[M]}\Pr(C=j|\mathbf{X})\right] \\
&= P_e,
\end{aligned}
\tag{16}
$$

where the second equality follows from (3), and the last equality is due to (2). This shows that $\psi_{\text{soft}}$ in (4) is an unbiased estimator of $P_e$.

Next, let $U_k \overset{i.i.d.}{\sim} Y_{M:M}$ for all $k \in [n]$, and note that $U_k \in \left[\frac{1}{M}, 1\right] := \mathcal{U}$. Define a function $f : \mathcal{U}^n \to \left[0, 1 - \frac{1}{M}\right]$ as $f(U_1, \ldots, U_n) := \frac{1}{n}\sum_{k=1}^{n}(1 - U_k)$, which is equivalent to $\psi_{\text{soft}}$ in (4). Then, for any $u_k \in \mathcal{U}$ and for all $k \in [n]$, we have that

$$
\begin{aligned}
&\sup_{v\in\mathcal{U}}|f(u_1,\ldots,u_{i-1},u_i,u_{i+1},\ldots,u_n) - f(u_1,\ldots,u_{i-1},v,u_{i+1},\ldots,u_n)| \\
&= \sup_{v\in\mathcal{U}}\left|\frac{1}{n}\left((1-u_i)-(1-v)\right)\right| \\
&\leq \frac{1}{n}\left(1 - \frac{1}{M}\right) := c_i.
\end{aligned}
\tag{17}
$$

By using McDiarmid's inequality [23], together with (16) and (17), we arrive at

$$
\begin{aligned}
\Pr\left(|\psi_{\text{soft}} - P_e| \geq \varepsilon\right) &= \Pr\left(|f(U_1,\ldots,U_n) - \mathbb{E}[f(U_1,\ldots,U_n)]| \geq \varepsilon\right) \\
&\leq 2\exp\left(-\frac{2n\varepsilon^2}{\left(1 - \frac{1}{M}\right)^2}\right).
\end{aligned}
\tag{18}
$$

By taking $n \to \infty$, we obtain

$$
\lim_{n\to\infty}\Pr\left(|\psi_{\text{soft}} - P_e| \geq \varepsilon\right) \leq \lim_{n\to\infty} 2\exp\left(-\frac{2n\varepsilon^2}{\left(1 - \frac{1}{M}\right)^2}\right) = 0,
\tag{19}
$$

which shows that $\psi_{\text{soft}}$ in (4) is a consistent estimator of $P_e$.

By setting $\varepsilon = \sqrt{\frac{\left(1-\frac{1}{M}\right)^2}{2n}\ln\frac{2}{\delta}}$ in (18) where $\delta \in (0, 1)$, we obtain

$$
\Pr\left(|\psi_{\text{soft}} - P_e| < \sqrt{\frac{\left(1-\frac{1}{M}\right)^2}{2n}\ln\frac{2}{\delta}}\right) > 1 - \delta.
\tag{20}
$$

This shows the convergence rate.

The estimator $\psi_{\text{soft}}$ in (4) can be viewed as the sample mean of $1 - \max_{j\in[M]} y_j$ in $\mathcal{D}$. Since $\max_{j\in[M]} y_j \sim Y_{M:M}$, from the central limit theorem, as $n \to \infty$, we obtain that

$$
\begin{aligned}
\sqrt{n}\left(\psi_{\text{soft}} - P_e\right) &\to \mathcal{N}(0, \text{Var}(1 - Y_{M:M})) \\
&= \mathcal{N}(0, \text{Var}(Y_{M:M})).
\end{aligned}
\tag{21}
$$

This concludes the proof of Theorem 1. □

## B.2 Proof of Proposition 1

*Proof.* We have that $Y_{M:M} \sim \max_{j \in [M]} Y_j$. Then, by letting $U_k \overset{i.i.d.}{\sim} Y_{M:M}$ for all $k \in [n]$, we obtain

$$
\begin{aligned}
\mathrm{Var}(\psi_{\mathrm{soft}}) &= \mathrm{Var}\left(\frac{1}{n}\sum_{k=1}^{n}(1 - U_k)\right) \\
&= \frac{1}{n}\mathrm{Var}(1 - U_k) \\
&= \frac{1}{n}\mathrm{Var}(U_k) \\
&= \frac{1}{n}\mathrm{Var}(Y_{M:M}).
\end{aligned} \tag{22}
$$

Moreover, by using the Bhatia-Davis inequality [7], we obtain $\mathrm{Var}(Y_{M:M}) = \mathrm{Var}(1 - Y_{M:M}) \le \left(1 - \frac{1}{M} - P_e\right)P_e$ since $\mathbb{E}[\psi_{\mathrm{soft}}] = P_e$, which gives

$$
\mathrm{Var}(\psi_{\mathrm{soft}}) \le \frac{\left(1 - \frac{1}{M}\right)P_e - P_e^2}{n}. \tag{23}
$$

Finally, we can maximize the term $\left(1 - \frac{1}{M} - P_e\right)P_e$ over $0 \le P_e \le 1 - \frac{1}{M}$. The maximum is indeed $\frac{\left(1 - \frac{1}{M}\right)^2}{4}$ attained by $P_e = \frac{1}{2}\left(1 - \frac{1}{M}\right)$. Hence,

$$
\begin{aligned}
\mathrm{Var}(\psi_{\mathrm{soft}}) &\le \frac{\left(1 - \frac{1}{M}\right)P_e - P_e^2}{n} \\
&\le \frac{\left(1 - \frac{1}{M}\right)^2}{4n},
\end{aligned} \tag{24}
$$

which concludes the proof of Proposition 1. $\qquad\square$

## B.3 Proof of Theorem 2

*Proof.* We start by noting that, by viewing $\psi_{\mathrm{soft}}$ in (4) as a sample mean of $(1 - Y_{M:M})$, the estimator $\mathsf{MoB}_K(\psi_{\mathrm{soft}}, \mathcal{D})$ is a median-of-means estimator. It has been shown in [62] that, provided that $\mathbb{E}[|1 - Y_{M:M} - P_e|^3] < \infty$, a median-of-means estimator satisfies

$$
|\mathsf{MoB}_K(\psi_{\mathrm{soft}}, \mathcal{D}) - P_e| \le 3(\mathrm{Var}(Y_{M:M}))^{\frac{1}{2}}\left(\frac{\mathbb{E}[|1 - Y_{M:M} - P_e|^3]}{\mathrm{Var}(Y_{M:M})^{\frac{3}{2}}}\frac{K}{n-K} + \sqrt{\frac{s}{n-K}}\right), \tag{25}
$$

with probability at least $1 - 4e^{-2s}$, for all $s \lesssim K$. From [54, eq. (3.8)], since $(1 - Y_{M:M}) \in \left[0, 1 - \frac{1}{M}\right]$ we have that

$$
\begin{aligned}
\mathbb{E}[|1 - Y_{M:M} - P_e|^3] &\le P_e\left(1 - \frac{1}{M} - P_e\right)\left(1 - \frac{1}{M} - 2P_e\right) \\
&\le \frac{\left(1 - \frac{1}{M}\right)^3}{6\sqrt{3}},
\end{aligned} \tag{26}
$$

where the last inequality follows by maximizing $P_e\left(1 - \frac{1}{M} - P_e\right)\left(1 - \frac{1}{M} - 2P_e\right)$ with respect to $0 \le P_e \le 1 - \frac{1}{M}$. Substituting (26) into (25) leads to

$$
\begin{aligned}
|\mathsf{MoB}_K(\psi_{\mathrm{soft}}, \mathcal{D}) - P_e| &\le 3(\mathrm{Var}(Y_{M:M}))^{\frac{1}{2}}\left(\frac{\left(1 - \frac{1}{M}\right)^3}{6\sqrt{3}(\mathrm{Var}(Y_{M:M}))^{\frac{3}{2}}}\frac{K}{n-K} + \sqrt{\frac{s}{n-K}}\right) \\
&= \left(\frac{\left(1 - \frac{1}{M}\right)^3}{2\sqrt{3}\mathrm{Var}(Y_{M:M})}\frac{K}{\sqrt{n-K}} + 3\sqrt{s\mathrm{Var}(Y_{M:M})}\right)\sqrt{\frac{1}{n-K}}.
\end{aligned} \tag{27}
$$

Under an appropriate choice of $n$ and $K$ (e.g., $K = o(n)$), the upper bound in (27) converges to 0, which proves the consistency of $\mathsf{MoB}_K(\psi_{\mathrm{soft}}, \mathcal{D})$.

For the breakdown point of $\mathsf{MoB}_K(\psi_{\text{soft}}, \mathcal{D})$, we start by observing that, if we have $\lfloor \frac{K+1}{2} \rfloor$ outliers, then at most $\lfloor \frac{K+1}{2} \rfloor$ base estimators $\psi_{\text{soft}}(\mathcal{D}_k)$'s in (6) can be affected by these outliers, i.e., they can be bad. In this case, $\mathsf{MoB}_K(\psi_{\text{soft}}, \mathcal{D})$ gives a bad estimate since more than half of the base estimators $\psi_{\text{soft}}(\mathcal{D}_k)$'s are bad. However, if the number of outliers is smaller than $\lfloor \frac{K+1}{2} \rfloor$, e.g., $\lfloor \frac{K+1}{2} \rfloor - 1$, then the median of $\psi_{\text{soft}}(\mathcal{D}_k)$, $k \in [K]$ is not a bad estimate since the number of outliers is smaller than half of the base estimators. Thus, from Definition 3 with $\tau = \lfloor \frac{K+1}{2} \rfloor$ and $\kappa + \tau = n$, we obtain $B\left(\mathsf{MoB}_K(\psi_{\text{soft}}, \mathcal{D})\right) = \lfloor \frac{K+1}{2} \rfloor \frac{1}{n}$.

The proof of the asymptotic normality of $\mathsf{MoB}_K(\psi_{\text{soft}}, \mathcal{D})$ follows directly from [62, Theorem 4]. In particular, if $K \to \infty$ and $K = o(\sqrt{n})$ as $n \to \infty$, it holds that

$$\sqrt{n}\left(\mathsf{MoB}_K(\psi_{\text{soft}}, \mathcal{D}) - P_e\right) \xrightarrow{d} \mathcal{N}\left(0, \frac{\pi}{2} \text{Var}(Y_{M:M})\right). \tag{28}$$

This concludes the proof of Theorem 2. $\qquad\square$

### B.4   Random permutation noise

One of the most common type of label noises is a noise that randomly permutes/shuffles the data labels [27], i.e., we have $(\boldsymbol{x}, \tilde{\boldsymbol{y}}) \in \widetilde{\mathcal{D}}_{\text{P}}$ where $\widetilde{\mathcal{D}}_{\text{P}}$ denotes the dataset with randomly permuted labels. This noise model can be categorized into two types [27], namely *instance-independent* and *instance-dependent*. The former perturbs the label $\boldsymbol{y}$ with a random permutation according to some probability distribution (the extreme noise case, namely unlabeled data or data without correspondence, where the label is permuted with probability one using a random permutation matrix, has also been studied extensively [36, 71, 73, 79, 87, 95]), which is independent of the input features. For example, a corruption by an instance-independent noise to $\boldsymbol{y} = [0.1, 0.7, 0.2]^\top$ may result in $\tilde{\boldsymbol{y}} = [0.1, 0.2, 0.7]^\top$. The other type of noise, the instance-dependent noise, flips/permutes the labels depending on the instance. For example, the label "car" is more likely to be flipped to "truck" rather than "cat".

We note that $\psi_{\text{soft}}$ in (4) only depends on the maximum value of $\boldsymbol{y}$, which does not change after that a random permutation is applied on $\boldsymbol{y}$. This property ensures that $\psi_{\text{soft}}$ is still an effective estimator of $P_e$ even when the labels are randomly permuted, as stated in the next theorem.

**Theorem 4.** *Let* $\widetilde{\mathcal{D}}_{\text{P}} = \{(\boldsymbol{x}_i, \tilde{\boldsymbol{y}}_i)\}_{i=1}^n$ *be a dataset that consists of noisy soft labels* $\tilde{\boldsymbol{y}}_i = P_i \boldsymbol{y}_i$ *where* $P_i, i \in [M]$ *is any random permutation matrix of size* $M \times M$. *Then,* $\psi_{\text{soft}}(\widetilde{\mathcal{D}}_{\text{P}})$ *satisfies all of the properties in Theorem 1 and Proposition 1.*

*Proof.* Since $\max_{j \in [M]} y_j$ is permutation-invariant with respect to $\boldsymbol{y}$, it follows that

$$\begin{aligned}
\psi_{\text{soft}}(\{(\boldsymbol{x}_i, \tilde{\boldsymbol{y}}_i)\}_{i=1}^n) &= \psi_{\text{soft}}(\{(\boldsymbol{x}_i, P_i \boldsymbol{y}_i)\}_{i=1}^n) \\
&= \psi_{\text{soft}}(\{(\boldsymbol{x}_i, \boldsymbol{y}_i)\}_{i=1}^n),
\end{aligned} \tag{29}$$

where the first equality follows since $\tilde{\boldsymbol{y}}_i = P_i \boldsymbol{y}_i$, and the second equality is due to the permutation-invariance property of the $\max$ function. The results in Theorem 1 and Proposition 1 then prove Theorem 4. $\qquad\square$

### B.5   Proof of Theorem 3

*Proof.* Without loss of generality, we let $\mathcal{X} = \{\boldsymbol{u}_1, \boldsymbol{u}_2, \dots, \boldsymbol{u}_K\}$, where $K = |\mathcal{X}|$ is the cardinality of $\mathcal{X}$, and $\boldsymbol{u}_k$ is the unique feature vector (or tensor). We denote by $\boldsymbol{v}_k \in \mathcal{Y}$ the soft label corresponding to $\boldsymbol{u}_k$, i.e., $\boldsymbol{v}_k = [\Pr(C = 1 | \mathbf{X} = \boldsymbol{u}_k), \dots, \Pr(C = M | \mathbf{X} = \boldsymbol{u}_k)]^\top$. With these definitions, we observe that due to the law of large numbers, we have

$$\begin{aligned}
\lim_{n \to \infty} \boldsymbol{s}(\boldsymbol{u}_k) &= \mathbb{E}[\widetilde{\mathbf{Y}} | \mathbf{X} = \boldsymbol{u}_k] \\
&\overset{\text{(a)}}{=} \mathbb{E}[\mathbb{E}[\widetilde{\mathbf{Y}} | \mathbf{Y}] | \mathbf{X} = \boldsymbol{u}_k] \\
&\overset{\text{(b)}}{=} \mathbb{E}[\mathbf{Y} | \mathbf{X} = \boldsymbol{u}_k] \\
&= \boldsymbol{v}_k,
\end{aligned} \tag{30}$$

where (a) follows from the law of total expectation and (b) is due to the fact that $\mathbb{E}[\widetilde{\mathbf{Y}}|\mathbf{Y}] = \mathbf{Y}$ since the noise is zero mean. Now, since for any $i \in [n]$ there exists a $k \in [K]$ such that $\boldsymbol{x}_i = \boldsymbol{u}_k$, we can write that

$$
\begin{aligned}
\lim_{n\to\infty} \mathsf{idx}(\boldsymbol{x}_i) &= \lim_{n\to\infty} \mathsf{idx}(\boldsymbol{u}_k) \\
&\stackrel{(a)}{=} \arg\max_{j\in[M]} \{(\boldsymbol{v}_k)_j\} \\
&= \arg\max_{j\in[M]} \{\Pr(C = j|\mathbf{X} = \boldsymbol{u}_k)\} \\
&:= \widehat{\mathsf{idx}}(\boldsymbol{u}_k),
\end{aligned}
\tag{31}
$$

where the equality in (a) follows from (30). Then, we can write $\psi_{\mathrm{DN}}(\widetilde{\mathcal{D}}_{\mathrm{A}})$ in (12) as

$$
\psi_{\mathrm{DN}}(\widetilde{\mathcal{D}}_{\mathrm{A}}) = \sum_{k=1}^{K} \frac{1}{n} \sum_{i=1}^{n} \mathbb{1}\{\boldsymbol{x}_i = \boldsymbol{u}_k\}(1 - (\widetilde{\boldsymbol{y}}_i)_{\mathsf{idx}(\boldsymbol{x}_i)}).
\tag{32}
$$

When $n \to \infty$, it follows that

$$
\begin{aligned}
\lim_{n\to\infty} \psi_{\mathrm{DN}}(\widetilde{\mathcal{D}}_{\mathrm{A}}) &\stackrel{(a)}{=} \sum_{k=1}^{K} \mathbb{E}\left[\mathbb{1}\{\mathbf{X} = \boldsymbol{u}_k\}\left(1 - \widetilde{\mathbf{Y}}_{\widehat{\mathsf{idx}}(\boldsymbol{u}_k)}\right)\right] \\
&\stackrel{(b)}{=} \sum_{k=1}^{K} \mathbb{E}\left[\mathbb{1}\{\mathbf{X} = \boldsymbol{u}_k\}\left(1 - \mathbb{E}[\widetilde{\mathbf{Y}}|\mathbf{Y}]_{\widehat{\mathsf{idx}}(\boldsymbol{u}_k)}\right)\right] \\
&\stackrel{(c)}{=} \sum_{k=1}^{K} \mathbb{E}\left[\mathbb{1}\{\mathbf{X} = \boldsymbol{u}_k\}\left(1 - \mathbf{Y}_{\widehat{\mathsf{idx}}(\boldsymbol{u}_k)}\right)\right] \\
&= \sum_{k=1}^{K} \mathbb{E}\left[\mathbb{1}\{\mathbf{X} = \boldsymbol{u}_k\}\left(1 - \max_{j\in[M]} \Pr(C = j|\mathbf{X} = \boldsymbol{u}_k)\right)\right] \\
&= \sum_{k=1}^{K} \mathbb{E}\left[1 - \max_{j\in[M]} \Pr(C = j|\mathbf{X} = \boldsymbol{u}_k) \,\Big|\, \mathbf{X} = \boldsymbol{u}_k\right] \Pr(\mathbf{X} = \boldsymbol{u}_k) \\
&= \mathbb{E}\left[1 - \max_{j\in[M]} \Pr(C = j|\mathbf{X})\right] \\
&= P_e,
\end{aligned}
\tag{33}
$$

where the labeled equalities follow from: (a) the law of large numbers, the assumption that $(\boldsymbol{x}_i, \widetilde{\boldsymbol{y}}_i) \sim P_{\mathbf{X},\widetilde{\mathbf{Y}}}$, and using (31); (b) using the law of total expectation; and (c) the fact that $\mathbb{E}[\widetilde{\mathbf{Y}}|\mathbf{Y}] = \mathbf{Y}$ since the noise is zero mean. This proves the consistency of $\psi_{\mathrm{DN}}$.

The asymptotic unbiasedness directly follows from the consistency together with the assumption of bounded support for the noise. Specifically, since $Z_j \in [a, b]$ for all $j \in [M]$, we have that

$$
\begin{aligned}
\lim_{n\to\infty} \mathbb{E}\left[\psi_{\mathrm{DN}}(\widetilde{\mathcal{D}}_{\mathrm{A}})\right] &= \mathbb{E}\left[\lim_{n\to\infty} \psi_{\mathrm{DN}}(\widetilde{\mathcal{D}}_{\mathrm{A}})\right] \\
&= \mathbb{E}[P_e] \\
&= P_e,
\end{aligned}
\tag{34}
$$

where the first equality follows by the dominated convergence theorem that can be applied since $|\psi_{\mathrm{DN}}(\widetilde{\mathcal{D}}_{\mathrm{A}})| \leq \max\{|a|, |1 - \frac{1}{M} + b|\}$, where $a$ and $b$ are the minimum and the maximum value in the support of the noise, respectively, and they are both finite. This shows the asymptotic unbiasedness of $\psi_{\mathrm{DN}}(\widetilde{\mathcal{D}}_{\mathrm{A}})$.

We are left to prove the denoising consistency. The denoised label $s(\boldsymbol{x}_i)$ in (11) is an unbiased estimator of $\boldsymbol{y}_i$ since

$$
\begin{aligned}
\mathbb{E}[\boldsymbol{s}(\mathbf{X}_i)] &\overset{\text{(a)}}{=} \mathbb{E}\left[\frac{\sum_{j=1}^n \mathbb{1}\{\mathbf{X}_i = \mathbf{X}_j\}\widetilde{\mathbf{Y}}_j}{\sum_{j=1}^n \mathbb{1}\{\mathbf{X}_i = \mathbf{X}_j\}}\right] \\
&\overset{\text{(b)}}{=} \mathbb{E}\left[\mathbb{E}\left[\frac{\sum_{j=1}^n \mathbb{1}\{\mathbf{X}_i = \mathbf{X}_j\}\widetilde{\mathbf{Y}}_j}{\sum_{j=1}^n \mathbb{1}\{\mathbf{X}_i = \mathbf{X}_j\}} \ \middle| \ N, \mathbf{X}_i\right]\right] \\
&\overset{\text{(c)}}{=} \mathbb{E}\left[\frac{1}{N}\sum_{k=1}^N \mathbb{E}\left[\widetilde{\mathbf{Y}}_k \ \middle| \ N, \mathbf{X}_i\right]\right] \\
&\overset{\text{(d)}}{=} \mathbb{E}\left[\frac{1}{N}\sum_{k=1}^N \mathbb{E}\left[\mathbf{Y}_i \mid \mathbf{X}_i\right]\right] \\
&= \mathbb{E}[\mathbf{Y}_i | \mathbf{X}_i],
\end{aligned}
\tag{35}
$$

where the labeled equalities follow from: (a) letting $\mathbf{X}_j$, $\mathbf{Y}_j$, and $\widetilde{\mathbf{Y}}_j$ for all $j \in [n]$ be independent copies of $\mathbf{X}$, $\mathbf{Y}$, and $\widetilde{\mathbf{Y}}$, respectively; (b) introducing a random variable $N \in [n]$ that counts the number of $\mathbf{X}_j$ such that $\mathbf{X}_j = \mathbf{X}_i$ and using the law of total expectation; (c) re-indexing $\widetilde{\mathbf{Y}}_k$, $k \in [N]$ for the $N$ pairs $(\mathbf{X}_j, \widetilde{\mathbf{Y}}_k)$ such that $\mathbf{X}_i = \mathbf{X}_j$; and (d) the fact that the noise is zero mean.

Note that $\boldsymbol{s}(\boldsymbol{x}_i)$ is the average of $n_{\boldsymbol{x}_i}$ noisy labels $\tilde{\boldsymbol{y}}$'s corresponding to $\boldsymbol{x}_i$. By Hoeffding's inequality, we then obtain

$$
\Pr\left(|\boldsymbol{s}(\boldsymbol{x}_i) - \boldsymbol{y}_i| \geq \varepsilon\right) \leq 2\exp\left(-\frac{2n_{\boldsymbol{x}_i}\varepsilon^2}{\left(1 - \frac{1}{M} - a + b\right)^2}\right).
\tag{36}
$$

Setting $\varepsilon = \sqrt{\frac{\left(1 - \frac{1}{M} - a + b\right)^2}{2n_{\boldsymbol{x}_i}} \ln \frac{2}{\delta}}$ leads to

$$
\Pr\left(|\boldsymbol{s}(\boldsymbol{x}_i) - \boldsymbol{y}_i| < \sqrt{\frac{\left(1 - \frac{1}{M} - a + b\right)^2}{2n_{\boldsymbol{x}_i}} \ln \frac{2}{\delta}}\right) > 1 - \delta.
\tag{37}
$$

This concludes the proof of Theorem 3. $\qquad\square$

## B.6 Properties of $\psi_{\text{genie}}(\widetilde{\mathcal{D}}_{\text{A}})$

The following proposition defines $\psi_{\text{genie}}$ and provides some properties of it.

**Proposition 2.** *Let $\widetilde{\mathcal{D}}_{\text{A}} = \{(\boldsymbol{x}_i, \tilde{\boldsymbol{y}}_i)\}_{i=1}^n$ be a dataset that consists of noisy soft labels $\tilde{\boldsymbol{y}}_i = \boldsymbol{y}_i + \boldsymbol{z}_i$ with $\boldsymbol{z}_i \overset{i.i.d.}{\sim} P_{\mathbf{Z}}$ such that $\mathbb{E}[\mathbf{Z}] = \mathbf{0}$. Let $c_i = \arg\max_{j \in [M]}(\boldsymbol{y}_i)_j$. Then, the following estimator*

$$
\psi_{\text{genie}}(\widetilde{\mathcal{D}}_{\text{A}}) = \frac{1}{n}\sum_{i=1}^n (1 - (\tilde{\boldsymbol{y}}_i)_{c_i})
\tag{38}
$$

*satisfies the following properties:*

1. *(Unbiasedness): It is an unbiased estimator of the BER;*

2. *(Consistency): It is a consistent estimator of the BER;*

3. *(Variance): $Var\left(\psi_{\text{genie}}(\widetilde{\mathcal{D}}_{\text{A}})\right) = \frac{Var(Y_{M:M}) + Var(Z)}{n}$;*

4. *(Asymptotic Normality): $\sqrt{n}(\psi_{\text{genie}}(\widetilde{\mathcal{D}}_{\text{A}}) - P_e) \overset{d}{\to} \mathcal{N}(0, Var(Y_{M:M}) + Var(Z))$ as $n \to \infty$.*

*Proof.* Define the random variable $C = \arg\max_{j\in[M]} Y_j$ that indicates the index of $\mathbf{Y}$ having the largest value, i.e., $Y_C = \max_{j\in[M]} Y_j$. We have that

$$
\begin{aligned}
\mathbb{E}[\psi_{\text{genie}}(\widetilde{\mathcal{D}}_{\text{A}})] &= \frac{1}{n}\sum_{i=1}^{n}\mathbb{E}[1 - \widetilde{\mathbf{Y}}_C] \\
&\overset{(a)}{=} \mathbb{E}[\mathbb{E}[1 - \widetilde{\mathbf{Y}}_C|\mathbf{Y}]] \\
&\overset{(b)}{=} \mathbb{E}[1 - \mathbf{Y}_C] \\
&= \mathbb{E}\left[1 - \max_{j\in[M]} Y_j\right] \\
&= P_e,
\end{aligned}
\tag{39}
$$

where the labeled equalities follow from: (a) the law of total expectation; and (b) the fact that $\mathbb{E}[\widetilde{\mathbf{Y}}|\mathbf{Y}] = \mathbf{Y}$ since $\mathbb{E}[\mathbf{Z}] = \mathbf{0}$. This shows the unbiasedness of $\psi_{\text{genie}}(\widetilde{\mathcal{D}}_{\text{A}})$.

Now, we show the consistency of $\psi_{\text{genie}}(\widetilde{\mathcal{D}}_{\text{A}})$. Due to the law of large numbers, as $n \to \infty$, we obtain that

$$
\begin{aligned}
\lim_{n\to\infty}\psi_{\text{genie}}(\widetilde{\mathcal{D}}_{\text{A}}) &= \lim_{n\to\infty}\frac{1}{n}\sum_{i=1}^{n}(1 - (\tilde{\mathbf{y}}_i)_{c_i}) \\
&= \mathbb{E}[1 - \widetilde{\mathbf{Y}}_C] \\
&= \mathbb{E}[\mathbb{E}[1 - \widetilde{\mathbf{Y}}_C|\mathbf{Y}]] \\
&= \mathbb{E}[1 - \mathbf{Y}_C] \\
&= P_e,
\end{aligned}
\tag{40}
$$

which demonstrates the consistency of $\psi_{\text{genie}}(\widetilde{\mathcal{D}}_{\text{A}})$.

We now compute the variance of $\psi_{\text{genie}}(\widetilde{\mathcal{D}}_{\text{A}})$. We have that

$$
\begin{aligned}
\psi_{\text{genie}}(\widetilde{\mathcal{D}}_{\text{A}}) &= \frac{1}{n}\sum_{i=1}^{n}(1 - (\tilde{\mathbf{y}}_i)_{c_i}) \\
&= \frac{1}{n}\sum_{i=1}^{n}(1 - (\mathbf{y}_i)_{c_i} - (\mathbf{z}_i)_{c_i}) \\
&= \psi_{\text{soft}}(\mathcal{D}) - \frac{1}{n}\sum_{i=1}^{n}(\mathbf{z}_i)_{c_i}.
\end{aligned}
\tag{41}
$$

This yields

$$
\begin{aligned}
\text{Var}\left(\psi_{\text{genie}}(\widetilde{\mathcal{D}}_{\text{A}})\right) &= \text{Var}\left(\psi_{\text{soft}}(\mathcal{D}) - \frac{1}{n}\sum_{i=1}^{n}Z_{c_i}\right) \\
&\overset{(a)}{=} \text{Var}\left(\psi_{\text{soft}}(\mathcal{D})\right) + \frac{1}{n}\text{Var}\left(Z\right) \\
&\overset{(b)}{=} \frac{\text{Var}(Y_{M:M})}{n} + \frac{1}{n}\text{Var}\left(Z\right),
\end{aligned}
\tag{42}
$$

where (a) follows from the independence between $\mathbf{Y}$ and $\mathbf{Z}$, and the fact that $Z_j$'s (with $j \in [M]$) are i.i.d.; and (b) follows by Proposition 1.

Finally, to show the asymptotic normality, we observe that from (41), we have that

$$\psi_{\text{genie}}(\widetilde{\mathcal{D}}_A) = \frac{1}{n}\sum_{i=1}^n \left(1 - (\boldsymbol{y}_i)_{c_i} - (\boldsymbol{z}_i)_{c_i}\right)$$

$$= \frac{1}{n}\sum_{i=1}^n \left(1 - \max_{j \in [M]}(\boldsymbol{y}_i)_j - (\boldsymbol{z}_i)_{c_i}\right)$$

$$\stackrel{d}{=} \frac{1}{n}\sum_{i=1}^n \left(1 - \max_{j \in [M]}(\boldsymbol{y}_i)_j - (\boldsymbol{z}_i)_1\right), \tag{43}$$

where the last equality follows since $(\boldsymbol{z}_i)_k \stackrel{d}{=} (\boldsymbol{z}_i)_\ell$ for all $(k, \ell) \in [M]^2$. Since (43) is the sample mean of $n$ realizations from $1 - Y_{M:M} - Z$, the central limit theorem leads to

$$\sqrt{n}(\psi_{\text{genie}}(\widetilde{\mathcal{D}}_A) - P_e) \stackrel{d}{\to} \mathcal{N}(0, \text{Var}(1 - Y_{M:M} - Z))$$

$$\stackrel{d}{=} \mathcal{N}(0, \text{Var}(Y_{M:M}) + \text{Var}(Z)), \tag{44}$$

where the last equality is due to the independence between $Y_{M:M}$ and $Z$. This concludes the proof of Proposition 2. $\qquad\square$

## C   Details on existing BER estimators / Additional experiments

### C.1   Details on existing BER estimators

#### C.1.1   k-NN BER bounds [16]

Let $P_e^{\text{NN}}$ be the error rate of the 1-nearest neighbor (NN) classifier with $n$ samples. Then, for binary classification, it follows that for $n \to \infty$ [16],

$$\frac{1}{2}\left(1 - \sqrt{1 - 2P_e^{\text{NN}}}\right) \le P_e \le P_e^{\text{NN}}. \tag{45}$$

The above bounds were proved by showing the convergence of the conditional posterior probability of the nearest neighbor to the true conditional posterior probability. For details, we refer an interested reader to [16].

To generalize (45) to $M$-classification problems, it was shown that for $n \to \infty$ [16],

$$\frac{M-1}{M}\left(1 - \sqrt{1 - \frac{M}{M-1}P_e^{\text{NN}}}\right) \le P_e \le P_e^{\text{NN}}. \tag{46}$$

In our experiments, we chose $k = 1$ as this value provides the tightest bounds among larger values of $k$ [16, 72], and we plotted the upper and lower bounds defined in (46). In particular, in order to evaluate $P_e^{\text{NN}}$ on given data samples, we generated true one-hot labels for all data samples from their soft labels by taking the index having the maximum value in the soft labels.

#### C.1.2   Generalized Henze-Penrose (GHP) divergence BER bounds [75]

Consider a parameter $p \in (0, 1)$ and two distributions $f_0$ and $f_1$. Then, the Henze-Penrose (HP) divergence between $f_0$ and $f_1$ is defined as,

$$D_p(f_0, f_1) = \frac{1}{4p(1-p)}\left(\int \frac{(pf_0(\boldsymbol{x}) - (1-p)f_1(\boldsymbol{x}))^2}{pf_0(\boldsymbol{x}) + (1-p)f_1(\boldsymbol{x})}\mathrm{d}\boldsymbol{x} - (2p-1)^2\right). \tag{47}$$

An appealing feature of $D_p$ is that it can be estimated directly from the data using the generalized Friedman-Rafsky (FR) statistic [28, 75], which uses the Euclidean minimal spanning tree.

Then, for $f_0$ and $f_1$ with prior probabilities $p$ and $1 - p$, the BER is bounded as [6],

$$\frac{1}{2} - \frac{1}{2}\sqrt{u_p(f_0, f_1)} \le P_e \le \frac{1}{2} - \frac{1}{2}u_p(f_0, f_1), \tag{48}$$

where

$$u_p(f_0, f_1) = 4p(1-p)D_p(f_0, f_1) + (2p-1)^2, \tag{49}$$

and $D_p(f_0, f_1)$ can be estimated based on the minimal spanning tree [6].

To generalize the HP bounds in (48), consider an $M$-class classification problem with $p_1, \ldots, p_M$ as class prior probabilities and class conditional probability densities given by $f_k(\boldsymbol{x}) := f(\boldsymbol{x}|c = k), k = 1, \ldots, M$. Let $f^{(M)}(\boldsymbol{x}) := \sum_{k=1}^{M} p_k f_k(\boldsymbol{x})$. Then, the generalized Henze-Penrose (GHP)-integral is defined as,

$$\mathsf{GHP}^{(M)}(f_i, f_j) = \int_{\mathcal{S}} \frac{f_i(\boldsymbol{x}) f_j(\boldsymbol{x})}{f^{(M)}(\boldsymbol{x})} \mathrm{d}\boldsymbol{x}, \tag{50}$$

where $\mathcal{S}$ is the support of $f^{(M)}(\boldsymbol{x})$. Let $\delta_{i,j}^{(M)} := \int \frac{p_i p_j f_i(\boldsymbol{x}) f_j(\boldsymbol{x})}{f^{(M)}(\boldsymbol{x})} \mathrm{d}\boldsymbol{x}$. Then, it was shown in [75] that the BER upper bound is

$$P_e \leq 2 \sum_{i=1}^{M-1} \sum_{j=i+1}^{M} \delta_{i,j}^{(M)}, \tag{51}$$

and the BER lower bound is

$$P_e \geq \frac{M-1}{M} \left( 1 - \left( 1 - \frac{2M}{M-1} \sum_{i=1}^{M-1} \sum_{j=i+1}^{M} \delta_{i,j}^{(M)} \right)^{\frac{1}{2}} \right). \tag{52}$$

In our experiments, we plotted the GHP BER upper and lower bounds defined in (51) and (52). Similar to the $k$-NN BER bounds, we generated true one-hot labels for all data samples before evaluating the GHP divergence $D_p$. Computing the above bounds requires to evaluate $\delta_{i,j}^{(M)}$ and this can be done by using the generalized FR statistics (for more details, see [75, Section IV.]).

## C.2 Additional experiments

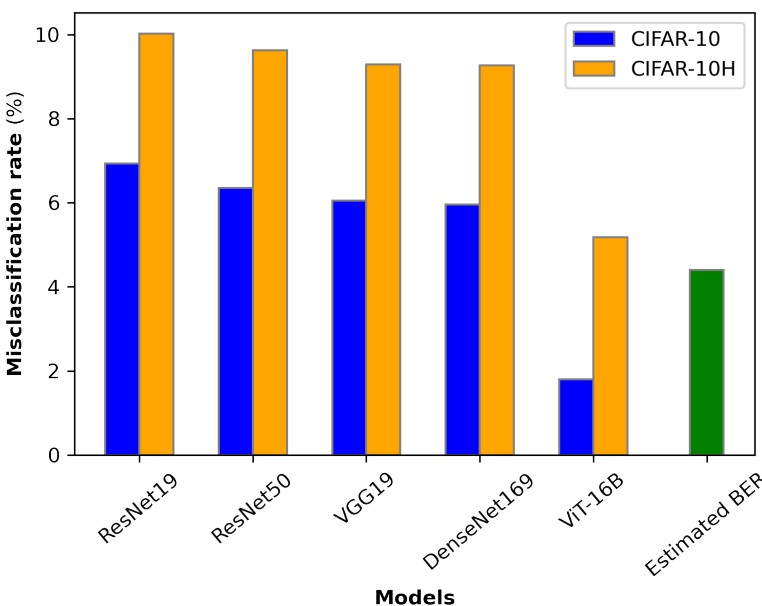

Figure 5: Misclassification error rates of several classifiers on CIFAR-10 and CIFAR-10H datasets.

In Figure 5, we compare the estimated BER with the misclassification error rate of a few popular neural network models on the CIFAR-10 and CIFAR-10H datasets. We chose ResNet [40], VGG [78],

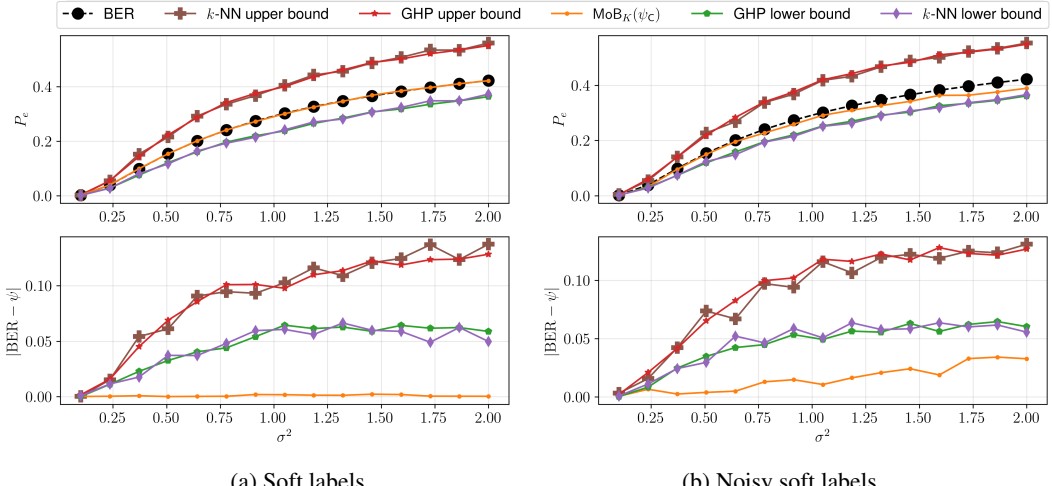

(a) Soft labels.              (b) Noisy soft labels.

Figure 6: Comparison of $\mathsf{MoB}_K(\psi_\mathsf{C})$ with the GHP bounds and the $k$-NN bounds with $k = 1$. The Gaussian samples are generated according to $\mathcal{N}(\boldsymbol{c}, \sigma^2 I_2)$, where $\boldsymbol{c} \in \{(1,1), (-1,1), (-1,-1), (1,-1)\}$ is equally likely. We consider $\mathbf{Z} \sim \mathcal{N}\left(\mathbf{0}, \frac{1}{5}I_4\right)$. $\mathsf{MoB}_K(\psi_\mathsf{C})$ uses $K = \lfloor\sqrt{n}\rfloor$, the Euclidean distance for d, and $r = 1/5$. We iterate the experiments 10 times and plot the average of them.

DenseNet [43], and Vision Transformer (ViT) [24] as classifiers to compare with and we trained these models on the CIFAR-10 dataset.[11] For the error rate evaluated on the CIFAR-10H dataset, we followed the same experiment setup as in [46], under which we generated the ground-truth labels for testset images from the soft labels (i.e., the probabilities for each class being associated with the input image) and evaluated the testset error rate. We iterated this procedure 20 times and plotted the average of the error rates. Figure 5 shows that the error rates on CIFAR-10 from the models are worse than the estimated BER, with the exception of ViT. As noted in [46], the reasons for having a lower error rate for ViT than the estimated BER are mainly due to 1) testset over-fitting and 2) various labeling difficulties on the dataset CIFAR-10H that is used for estimating the BER. After compensating for such difficulties on CIFAR-10H, all models' error rates (evaluated on CIFAR-10H) are worse than the estimated BER; this suggests that the estimated BER may be a good estimation since the error rate of any classifier is larger than the BER.

Figure 6 shows the comparison of $\mathsf{MoB}_K(\psi_\mathsf{C})$ with the GHP and $k$-NN bounds on the BER. For each class $\boldsymbol{c} \in \{(1,1), (-1,1), (-1,-1), (1,-1)\}$, we generated 500 Gaussian samples $\mathcal{N}(\boldsymbol{c}, \sigma^2 I_2)$ to have a total of 2,000 samples. The label noise was generated according to $\mathbf{Z} \sim \mathcal{N}(\mathbf{0}, \frac{1}{5}I_4)$. The case of no outliers is considered in this experiment. We observe that our BER estimator outperforms the other estimators over $\sigma^2 \in [0.1, 2]$ in both cases of noiseless soft labels and noisy soft labels.

We also made performance comparisons varying different parameters (i.e., $\mu, \sigma^2, r$) in Figures 7, 8, 9, and 10. Similar to the experiment setting for Figure 6, we generated a total of 2,000 Gaussian samples with different parameters that are specified in the caption of each figure. The effect of $\mu$ and $\sigma^2$ on the BER estimation is shown in Figures 7 and 8 under different choices of the radius $r$. We also provide performance comparisons of our estimator with the other estimators for different values of the noise power $\sigma^2_{\mathbf{Z}}$ in Figures 9 and 10. Choosing an appropriate value for $r$ is critical in the estimation of the BER as shown in Figure 10. An appropriate value for $r$ can be empirically determined by using the elbow method [84], which is commonly used for choosing the number of clusters in $k$-means clustering algorithms. Using Figure 10, we selected $r = 0.2$ (elbow point) for the main experiment in Figure 2. Overall, from these figures, we observe that our estimator outperforms the GHP and the 1-NN classifier BER bounds in various settings.

---

[11]We fine-tuned the official ViT-16B model (pre-trained on imagenet21k) on CIFAR-10 dataset; for this we used the implementation in [48]. Moreover, we trained the other models using the implementation in [70].

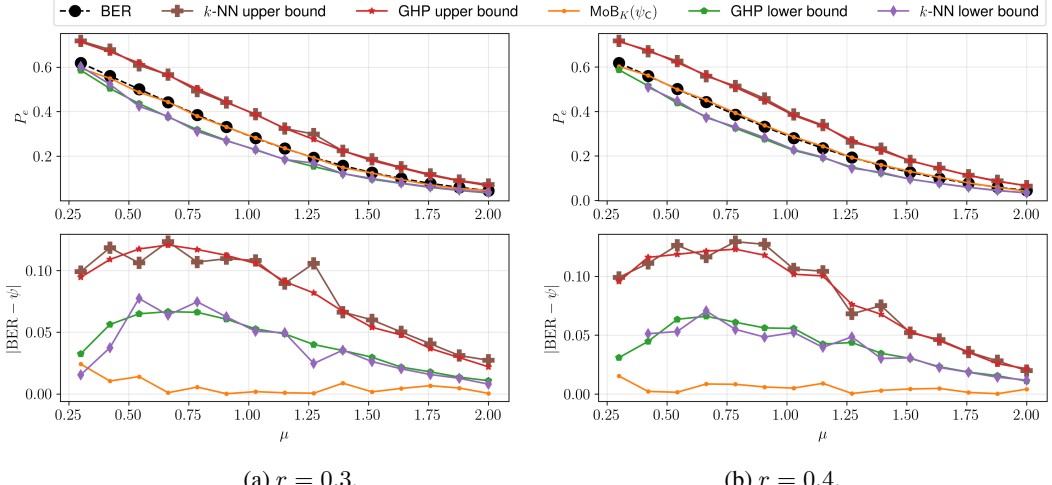

(a) $r = 0.3$.

(b) $r = 0.4$.

Figure 7: Comparison of $\mathsf{MoB}_K(\psi_\mathsf{C})$ with the GHP bounds and the $k$-NN bounds with $k = 1$. The Gaussian samples are generated according to $\mathcal{N}(\boldsymbol{c}, I_2)$, where $\boldsymbol{c} \in \{(\mu, \mu), (-\mu, \mu), (-\mu, -\mu), (\mu, -\mu)\}$ with equal probability. We consider $\mathbf{Z} \sim \mathcal{N}\left(\mathbf{0}, \frac{1}{5}I_4\right)$. $\mathsf{MoB}_K(\psi_\mathsf{C})$ uses $K = \lfloor\sqrt{n}\rfloor$, the Euclidean distance for d, and $r \in \{0.3, 0.4\}$. We iterate the experiments 10 times and plot the average of them.

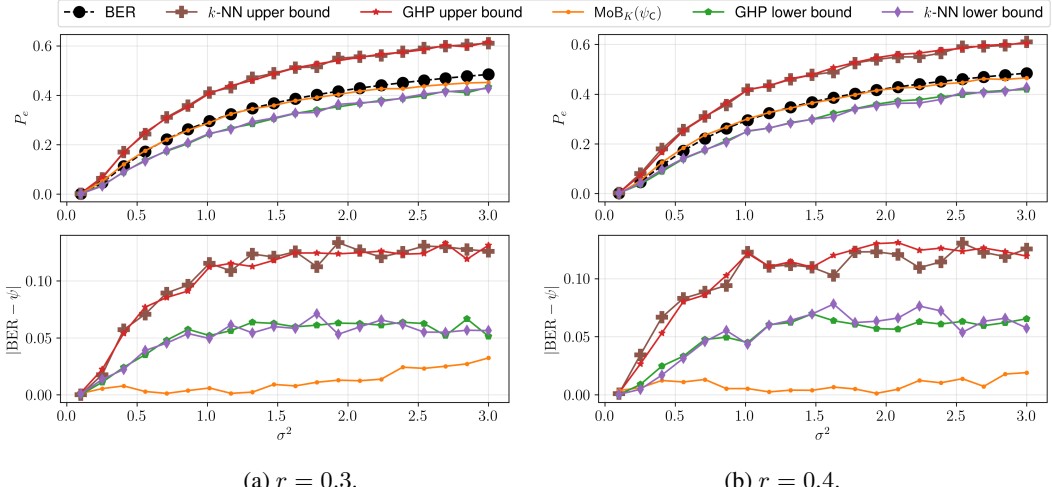

(a) $r = 0.3$.

(b) $r = 0.4$.

Figure 8: Comparison of $\mathsf{MoB}_K(\psi_\mathsf{C})$ with the GHP bounds and the $k$-NN bounds with $k = 1$. The Gaussian samples are generated according to $\mathcal{N}(\boldsymbol{c}, \sigma^2 I_2)$, where $\boldsymbol{c} \in \{(1, 1), (-1, 1), (-1, -1), (1, -1)\}$ with equal probability. We consider $\mathbf{Z} \sim \mathcal{N}\left(\mathbf{0}, \frac{1}{5}I_4\right)$. $\mathsf{MoB}_K(\psi_\mathsf{C})$ uses $K = \lfloor\sqrt{n}\rfloor$, the Euclidean distance for d, and $r \in \{0.3, 0.4\}$. We iterate the experiments 10 times and plot the average of them. As expected, a larger value of $r$ in Figure 8b yields better performance than a smaller value of $r$ in Figure 8a.

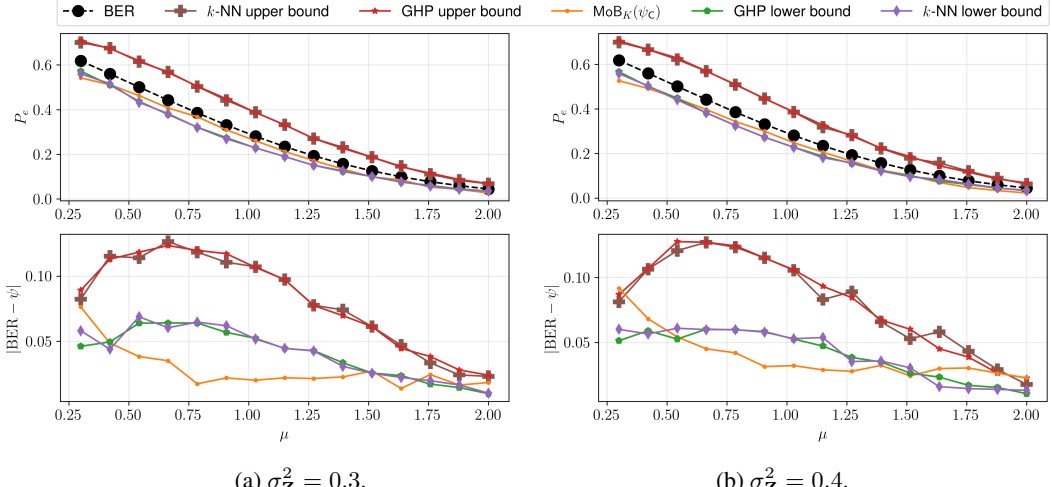

(a) $\sigma_{\mathbf{Z}}^2 = 0.3$.

(b) $\sigma_{\mathbf{Z}}^2 = 0.4$.

Figure 9: Comparison of $\mathsf{MoB}_K(\psi_\mathsf{C})$ with the GHP bounds and the $k$-NN bounds with $k = 1$. The Gaussian samples are generated according to $\mathcal{N}(\boldsymbol{c}, I_2)$, where $\boldsymbol{c} \in \{(\mu, \mu), (-\mu, \mu), (-\mu, -\mu), (\mu, -\mu)\}$ with equal probability. We consider $\mathbf{Z} \sim \mathcal{N}\left(\mathbf{0}, \sigma_{\mathbf{Z}}^2 I_4\right)$. $\mathsf{MoB}_K(\psi_\mathsf{C})$ uses $K = \lfloor\sqrt{n}\rfloor$, the Euclidean distance for d, and $r = 0.2$. We iterate the experiments 10 times and plot the average of them. In both figures, the performance of our BER estimator at the extreme values of $\mu$ is worse than the performance of the other estimators; this is due to an inappropriate choice of $r$.

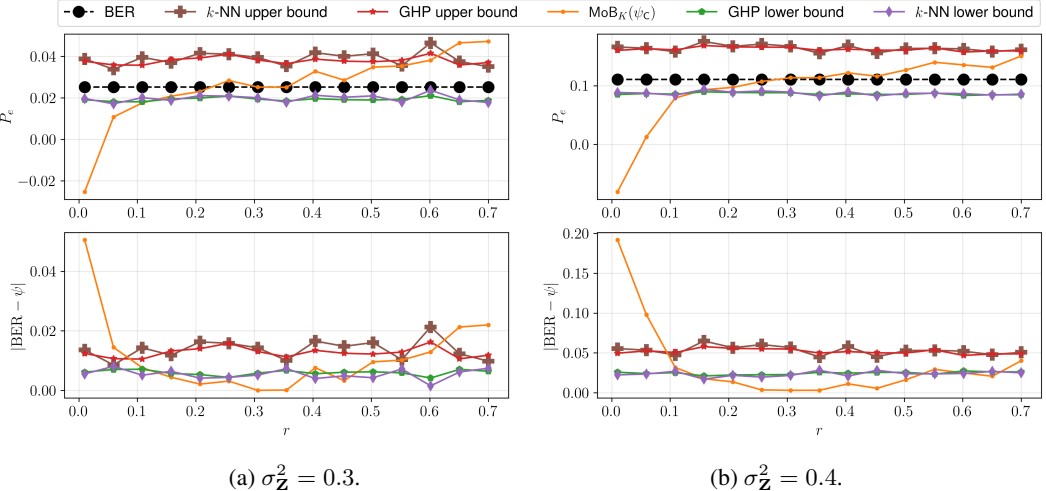

(a) $\sigma_{\mathbf{Z}}^2 = 0.3$.

(b) $\sigma_{\mathbf{Z}}^2 = 0.4$.

Figure 10: Comparison of $\mathsf{MoB}_K(\psi_\mathsf{C})$ with the GHP bounds and the $k$-NN bounds with $k = 1$ with respect to the cluster radius $r$. The Gaussian samples are generated according to $\mathcal{N}(\boldsymbol{c}, I_2)$, where $\boldsymbol{c} \in \{(1, 1), (-1, 1), (-1, -1), (1, -1)\}$. We consider $\mathbf{Z} \sim \mathcal{N}\left(\mathbf{0}, \sigma_{\mathbf{Z}}^2 I_4\right)$. $\mathsf{MoB}_K(\psi_\mathsf{C})$ uses $K = \lfloor\sqrt{n}\rfloor$ and the Euclidean distance for d. We iterate the experiments 10 times and plot the average of them.

