# OpenReview forum: "Demystifying the Optimal Performance of Multi-Class Classification"
_NeurIPS.cc/2023/Conference — NeurIPS 2023 poster_

### Official Review · Reviewer_HZ6f · 2023-07-03

**Soundness:** 3 good
**Presentation:** 2 fair
**Contribution:** 3 good
**Rating:** 6
**Confidence:** 4

**Summary:**

This paper tackles the important problem of performance estimation in a multi-class setting. The contributions come in the form of several Bayesian Error Rate estimators for different multi-class variants. Several theoretical guarantees are provided throughout the paper, as well as empirical results on synthetic and real dataset.

**Strengths:**

# Motivation
The problem of estimating and validating the performance of a classifier in a multi-class setting has been a hot topic in Machine Learning. As such, the results provided in this paper are of prime interest for the ML community. Estimating the BER can lead to a better understanding of the multi-class problem and to novel methods for such setting.

# Completness
The main strength of this paper resides in providing theoretical results for various settings. The first result is provided for the vanilla case of multi-class learning, where the data is drawn i.i.d. from a hidden distribution. The authors then provide a robust version of the previous result. Finally the paper provides several adaptation of the previous results to the noisy setting, i.e. when noise is added to the labels.

# Experimental results
The experimental section provides extensive empirical validation of the theoretical results. Several synthetic and real-world datasets are analyzed and discussed in Section 5 thus shedding light on the benefits of the proposed estimates.

**Weaknesses:**

# Clarity
While the main motivations are easy to follow, and the paper is overall very-well organized, in several occasions definitions and notations are poorly introduced. In lines 76-78, the definition of Y is misleading since it uses the same notation as in multi-class multi-label learning, particularly since no assumption is made on c and how it is retrieved from y. In line 79, it's not clear what $e$ in $P_e$ stands for and how it is related to the definition that follows.

In line 86, "an optimal classifier." implies that several classifiers can achieve BER. In other words, looking at the problem from an optimization problem point of view, it means that there are several global minima exist. I'm not quite sure I fully grasp the interest of such an assumption.

In line 111 the notation $Y_{M:M}$ is used without being properly introduced before or in the remainder of the paper.

In Definition 4, it's unclear wether an assumption on class distribution exists when defining the K subsets. The absence of such an assumption might explain the trade-off discussed in line 163, in which case it should be clearly noted in the paper.

I'm not quite sure I understand the role of paragraph Notation (lines 65 to 71), when most of the notations are loosely introduced in the remainder of the paper.

Equation (11) is quite puzzling. If I'm reading it right, the denoise estimator is based on samples that have the same exact $\bf x$, i.e. it implies that there are duplicated data in the learning dataset, which seems odd compared to the i.i.d. assumption of line 76. The denoise estimator of Equation (14) seems like a better and more realistic choice.

# Conclusion
While the experimental section provides several discussions on the proposed estimators, I strongly think that a Conclusion section is needed.

**Questions:**

* Can the framework be extended to multi-label classification?

**Limitations:**

Not applicable.

---

> ### Author Rebuttal · Authors · 2023-08-10
>
> We thank the Reviewer for their time and effort put into reading and reviewing our manuscript.
>
> Clarity 1. The focus of our paper is on multi-class single-label learning, where each data sample is associated with only one true class.
> Our rationale behind representing $\mathbf{y}$ as a vector stems from the fact that we consider soft labels and hence,  $\mathbf{y}$ provides the soft-valued information of each class (see eq.(3)).
> The Bayes error rate (BER) is denoted with $P_e$ (see Definition 1), where the subscript $e$ stands for error.
>
>
> Clarity 2. We define a classifier as optimal if it achieves the BER that is unique (i.e., there is only one global minimum).
> However, there may exist several classifiers that achieve the BER.
> For example, if two different classifiers (with different complexity or algorithm) achieve the BER, both would be considered as optimal.
> This is the reason for using the wording ``an optimal classifier''.
>
> Clarity 3. We defined $Y_{M:M}$ in the Notation (lines 69 and 70).  In particular, $Y_{M:M}$ is the $M$-th order statistic of $\mathbf{Y}$, that is,  the largest value of $\mathbf{Y}$.
>
> Clarity 4. Definition 4 is just a definition, which holds without any assumption on the class distribution. However, in Section 3, we consider the case of soft labels and hence, Theorem 2 holds under this assumption. In the revised version of the paper, we will add this assumption in the statement of Theorem 2.
>
> Clarity 5. We acknowledge that part of the notation introduced in lines 65-71 is repeated in the paper and may be familiar to an audience in the machine learning community. However, we believe that these few lines clearly introduce the reader to some notation that will be extensively used throughout the paper.
>
> Clarity 6. We made the assumption that the feature set $\mathcal{X}$ is finite, which implies the potential existence of duplicated data samples with a non-zero probability. However, this does not create any conflict with the i.i.d. assumption. An illustrative example is the i.i.d. sampling from the uniform distribution supported on {1, 2, 3}; each sample data can have one of these three values with equal probability and two different data samples can have the same value. As another example, in the MovieLens dataset in our experiment, two different users can give a different rating to the same movie.
>
> Conclusion. We thank the Reviewer for this suggestion on which we agree. We will make sure to add a Conclusion section in the revised paper.
>
> Question. This is a good question. Extending our framework to multi-label classification is a very interesting area for future work.
> To the best of our knowledge, several metrics (other than the BER) have been adopted for multi-label classification. Examples are the exact match ratio, the F1 score, and the Hamming loss. We believe that extending our framework to estimate the minimum Hamming loss of multi-label classification is not too challenging.
> However, extending it to estimate the other metrics may not be straightforward and may require the development of new tools.

---

> > ### Comment · Reviewer_HZ6f · 2023-08-15
> >
> > Thank you for your replies, I have a better understanding of the contribution. I'm changing my rating to 6, mostly due to the changes needed for the overall structure of the paper.

---

> > > ### Author Response · Authors · 2023-08-15
> > >
> > > We thank the Reviewer for reading through our rebuttal, and for changing their original score. The comments of the Reviewer greatly improve our paper.

---

### Official Review · Reviewer_iHwx · 2023-07-04

**Soundness:** 3 good
**Presentation:** 3 good
**Contribution:** 3 good
**Rating:** 6
**Confidence:** 4

**Summary:**

This paper studies the estimation of the Bayes error rate (BER) in the multiclass classification problem. BER is the best classification error (in terms of expectation) that can be achieved by the Bayes optimal classifier. First, this paper studies the soft labels case, where the direct extension of the estimator proposed by Ishida et al. [ICLR2023] can be shown to be valid in the multiclass case. Next, this paper studies the case where the soft labels are noisy and proposed the median-of-means estimator to improve the robustness. Furthermore, the denoise estimator cluster-based BER estimators are also proposed. Experiments are also provided to justify the usefulness of the proposed methods.

**Strengths:**

1. Writing is easy-to-follow: paper motivation, objectives, and solutions are clear.
2. Proposed methods are intuitive, easy to implement, and theoretically guaranteed. Many estimators are provided for different use cases.
3. I found the discussion about the robustness of the estimator $\psi_{\mathrm{soft}}$ interesting and the proposed median-of-BERs estimator to mitigate this problem is theoretically guaranteed and practically effective in experiments. In theory, I found the discussion in Lines 158-166 provides interesting comparisons between $\psi_{\mathrm{soft}}$ and its MoB estimator.
4. Experiments are convincing to show that the extension beyond straightforward estimator $\psi_\mathrm{soft}$ (MoB, cluster, denoising)can significantly outperform $\psi_\mathrm{soft}$.

**Weaknesses:**

1. Experiments to compare other methods may not be sufficient. In my understanding, only experiments with synthetic datasets with limited hyperparameter choice are conducted to justify the superior of the proposed methods (result in Figure 2). Additional experiments in Appendix C are also only synthetic datasets with the same hyperparameter choice as Figure 2.
2. Evaluation of the non-synthetic dataset is difficult to know if the Bayes error rate is well estimated, as also suggested in the paper that the labels themselves can be unreliable because of human errors (as discussed in Lines 312-314). But this issue is very problematic by itself and I have no idea how to better deal with this.
3. Soft labels can be restrictive in many scenarios which may make the proposed method prohibitive. Nevertheless the cluster estimator sounds in (15) reasonable as a good heuristic for BER estimation.


**Questions:**

1. Regarding Section 3.1, it is still not clear to me why having a higher value of the breakdown point guarantees more robustness of the estimator (Lines 139-141). What is the meaning of robust here? Does it mean a more accurate estimator under high noise?
2. I feel Definition 3 is quite difficult to interpret and more discussion could be useful for readers to grasp the importance and how to use Definition 3. I failed to fully appreciate it and would like to ask some questions.
- 2.1 What is definition of $\mathcal{D}^{(\kappa + \tau)}$?
- 2.2 How to interpret the meaning of Eq. (5)? Is there an intuitive explanation of the value of the breakdown points?
- 2.3 Why $B(\psi_\mathrm{soft}) = \frac{1}{n}$ is not robust? Why we can conclude that this makes the estimator not robust (Line 143)? What number of $B(\psi)$ is robust?
3. Questions about Figure 2 experiments:
- 3.1 I don't see $\psi_\mathrm{soft}$ but rather $MoB_K(\psi_c)$, is this a typo? If so, which methods are compared? I think it is also useful to see the performance of other proposed estimators as well.
- 3.2  Is it reasonable to use $k=1$ for $k-NN$ bound? I am not familiar with this method but I feel $1$-nearest neighbor may not be robust enough.
- 3.3 How hyperparameter selection is carried out? I feel the choice of $K=44$ for the proposed estimator is quite unintuitive and was wondering about the sensitivity of this hyperparameter.
4. Is there a theory to justify the statistical properties of the cluster estimator (15)? If not, is there a way to theoretically validate this estimator?
5. It is difficult to know the accuracy of BER estimation from Table 1. Is it possible to also put a reference of a model trained for these classification tasks similar to Figure 4 in Ishida et al?
6. It is discussed that MoB estimator is more robust and therefore gives lower value in Table 1. I was wondering if it is possible to have an outlier that has a low value also?

Typo:
1.Line 229: nosy -> noisy?
2. In some places $MoB$ is used instead of $MoB_K$, e.g., legend in Figure 2 and some other places. Maybe we should unify it if this is not intentional.

**Limitations:**

I have no further things to add here. Please see the weaknesses and questions section for some discussions on limitations.

---

> ### Author Rebuttal · Authors · 2023-08-10
>
> We thank the Reviewer for their thorough reading of our manuscript and their effort in carefully reviewing it.
>
> Weakness 1. We agree with the Reviewer that comparisons with SOTA BER bounds are performed only on synthetic datasets. However, this is not due to a limitation of our approach, but rather to the lack of SOTA BER estimators that are applicable to complex classification tasks.  SOTA BER estimators either require to find the data distribution/divergence or have a prohibitive complexity. Instead, we feel that our BER estimation methods are well-suited for such challenging tasks, as we have shown for the MovieLens dataset. That said, as the Reviewer has also suggested, we are training a model using the benchmark and real-world datasets in Table I. Unfortunately, we could not obtain the results by the time this rebuttal was due, but we will incorporate them in the revised version.
>
> Weakness 2. It is true that labels can be unreliable, and this is one of the major challenges in understanding if the BER is well estimated.  This is the main motivation behind our contribution of proposing denoising methods to reduce the noise that can corrupt the data labels.  To the best of our knowledge, this concept of leveraging denoising methods for BER estimation is new. Our results also suggest that this technique is effective and hence, we feel that it could be well-suited to mitigate the unreliability aspect of the data labels.
>
> Weakness 3. It is true that currently soft labels are less used compared to hard labels. However, we feel that the use of soft labels is growing considerably (see first paragraph in Section 3). In Section 4.2, we have also shown that the noisy soft label framework can be used to study the case of one-hot labels. This study leads to the BER estimator in eq.(15), which the Reviewer deems reasonable.
>
> Question 1. We consider robustness to outliers, where an outlier is a data sample that is corrupted by high noise. The breakdown point captures how robust a model is with respect to outliers. For example, the sample mean (i.e.,  an estimate of the mean of $X$, which is given by $\hat{\mu} = \frac{1}{n}\sum_{i=1}^n x_i$ where {$x_i$}$_{i=1}^n$ is a set of realizations of $X$) would fail to estimate the true mean even for the case when there is a single outlier (e.g., $x_i = \infty$ or an $x_i$ with a value very different from the true mean). Thus, the sample mean has a breakdown point equal to $\frac{1}{n}$, meaning that one outlier can break the estimate. Differently, the median is more robust since it has a breakdown point of $\frac{1}{2}$, i.e. half of the samples need to be outliers for breaking the estimate.
>
> Question 2.1. Per Definition 3, we have that $\mathcal{D}^{(\kappa + \tau)} =$ {$ D_1,\ldots,D_{\kappa+\tau} $}.
>
> Question 2.2. Eq.(5) quantifies the breakdown point for an estimator $\psi: \Omega^{\tau+\kappa}\to\Theta$. An estimator $\psi$ breaks if $\psi(\mathcal{D}^{(\kappa+\tau)})$ -- i.e., $\psi$ evaluated on the clean dataset with $\kappa + \tau$ data samples -- changes of at least $||\mathsf{rad}(\Theta)||$ when $\tau$ clean data samples are replaced with $\tau$ outliers. The breakdown point is defined as the minimum value of $\frac{\tau}{\tau+\kappa}$,  i.e., the ratio between the number of outliers ($\tau$)  and the total number of data samples ($\kappa + \tau$), such that $\tau$ outliers are sufficient to break $\psi$.
>
> Question 2.3. Per Definition 3, the breakdown point (see also our response to Question 2.2) is the minimum value of $\frac{\tau}{\tau+\kappa}$ such that $\tau$ outliers suffice to break the estimator $\psi$. In this sense $B(\psi_{{\rm{soft}}}) = \frac{1}{n}$ is not robust. In general, $\frac{1}{2}$ is the maximum value of a breakdown point.
>
> Question 3.1. This is a typo and we will fix it: in the caption, it should be $\mathsf{MoB}_K (\psi _{\mathsf{C}})$. Among the proposed estimators, in Figure 2 we only evaluated $\mathsf{MoB}_K(\psi _{\mathsf{C}})$. This is because we observed that it performs well with respect to SOTA methods and hence, we omitted the other estimators.
>
> Question 3.2. The optimal value of $k$ is not known and this might be one of the reasons for which generally, it is set to $1$. Moreover, the value of $k$ should be a small fraction of the total number of samples, and this could be another reason for setting it to $1$. That said, it may happen (and this is the case in the asymptotics) that increasing $k$ leads to a more accurate BER estimation. In the revised version, we will also provide the BER curve relative to the optimal value of $k$ (which we will find numerically).
>
> Question 3.2. We chose $K=\lfloor \sqrt{n} \rfloor$ based on Theorem 2, which shows that if $K<o(\sqrt{n})$ and $n\to\infty$, then $\mathsf{MoB}_{K}$ is asymptotically normal. Thus, we chose $K$ to be large enough (for robustness), yet smaller than $\sqrt{n}$.
>
> Question 4. We did not investigate the statistical properties of the cluster estimator in eq.(15); this is an interesting future direction. As Reviewer 2nax pointed out, this estimator resembles the Nadaraya-Watson estimator with the Parzen window kernel. Thus, we believe that a theoretical validation of the estimator in eq.(15) may be doable.
>
> Question 5. This is a great suggestion! We have started working on this, but we could not obtain the results by the time this rebuttal was due. We will incorporate these suggestions in the revised version.
>
> Question 6. In general, there might be outliers that lead to lower values of the estimated BER. However, since the labels belong to $[0,1]$ and the estimated BER values in our examples are very small ($0.004\sim0.04$), we believe that outliers with large values would be more impactful on the final estimate than outliers with small values. For instance, if the true estimate is around $0.05$, outliers with a smaller value than this might not be even considered as outliers.
>
> Typo 1 and Typo 2. We will fix these in the revised version of the paper.

---

> > ### Comment · Reviewer_iHwx · 2023-08-16
> > **Thank you very much for your kind explanation.**
> >
> > I have read the rebuttal and it made me understand much more about the part that I could not fully understand at first. Thank you! I raised my score to 6. In particular, I found the responses about breakdown points (Q1-2), low-value outliers (Q6), and how to choose K (Q3.2) very satisfactory and they are the main reasons I updated to score.
> >
> > Maybe it is just me but I found the explanation in the current paper was not that easy to understand about the breakdown point (as I am not familiar with it). I found the explanations in rebuttals for Q 1 and Q 2.2 highly useful to grasp this concept. It might be useful to include such explanations in the paper if possible.

---

> > > ### Author Response · Authors · 2023-08-16
> > >
> > > We thank the Reviewer for having read our rebuttal, and for raising their original score from 5 to 6. The comments of the Reviewer greatly helped improve our paper, and we will for sure include the explanations of the rebuttal in the revised version of the paper.

---

### Official Review · Reviewer_2nax · 2023-07-04

**Soundness:** 3 good
**Presentation:** 3 good
**Contribution:** 2 fair
**Rating:** 6
**Confidence:** 3

**Summary:**

This paper proposes a few methods for estimating Bayes error rates of multi-class classification in different scenarios. The proposed method generalize the previous soft-label approach for binary classification [Ishida et al., 2023] and also extends it for robustness to label noise and outliers. The first extension uses the median-of-means in place of the ordinary average for better robustness, and the second one uses the feature-wise average assuming the features are discrete. The third one uses a Nadaraya-Watson estimator with the Parzen kernel for incorporating data points near each point. The paper provides theoretical analyses for the estimation errors, convergence rates, and the robustness guarantees. Finally, the paper presents experiments using synthetic data and three benchmark datasets demonstrating the superiority of the proposed methods.

**Strengths:**

- The paper is well-written and pleasant to read.
- The extensions to the noisy cases including the one with hard (one-hot) labels may be practically useful because many datasets do not have soft labels.
- The use of the median of means to this problem seems interesting. The theoretical results about this part are also insightful.
- The experiments show the clear superiority of the proposed method over the baseline methods.

**Weaknesses:**

- The multi-class extension when soft labels are provided seems straightforward.
- The denoising method in Eq. (11) is a bit disappointing because it heavily relies on the assumption that the features are discrete.
- The denoising method in Definition 6 could be applied to any $\mathcal{X}$ with a distance metric, but this essentially estimates the conditional probability $P_{Y|X}$, which contradict the following claim: "This is an appealing feature of our work, different from taking a plug-in approach that first estimates the istribution from which the data is drawn, and then evaluates the BER. Indeed, our BER estimators, which are proved to be unbiased, consistent and robust to label noise and outliers, do not require the estimation of the data probability density to perform an effective BER estimation."
- Regarding the previous point, if the method in Definition 6 is no better than the SOTA method in the conditional probability estimation, the estimate will be an upper bound of the SOTA score (and the Bayes error). However, it is not convincing to say the proposed method, which looks like a classical method, is better than the SOTA method.

**Questions:**

### Major comments
- How does one choose $r$? More importantly, how did the authors choose it in the experiments?
- The proposed denoising method essentially estimates the class posterior probability using the kernel method. What is the motivation for using this method when there are many other classification methods performing well?
- The footnote on page 2 says "We assume that $\mathcal{X}$ is a finite set, but several of our results easily extend to the case when $\mathcal{X}$ is an infinite set." Having a finite $\mathcal{X}$ is very restrictive. Why not work on the infinite case in the first place if the extension is easy? What about the continuous case?

### Minor comments
- Line 137, "$\operatorname{rad}(\theta)$ is the vector consisting of $L$ radiuses along the main axes of the largest $L$-dimensional ellipsoid in replaced by outliers.": I could not understand this. What are the $L$ radiuses and the vector consisting of them, more precisely?
- Lines 164-165, "For example, setting $K = \ln n$ leads to a higher breakdown point (and hence, is more robust) than setting $K = \sqrt{n}$." Doesn't a larger $K$ have better robustness?
- Line 178, "We assume that z has i.i.d. components each with zero mean (without loss of generality)." I think this does lose the generality.
- In footnote 3 on page 5, is the definition of sub-Gaussianity correct?
- I am not very comfortable to call the method in Definition 6 a "nearest-neighbor" method because it uses a fixed kernel (with a fixed $r$). I would call it a Nadaraya-Watson estimator with the Parzen window kernel.
- I suggest using different letters for $P_e$ and $P_e(\cdot)$.

**Limitations:**

A limitation is that the paper assumes that the set of features is finite.

---

> ### Author Rebuttal · Authors · 2023-08-10
>
> We thank the Reviewer for their thorough reading of our manuscript, and for the positive assessment of our contribution.
>
> Weakness 1. We agree that the multi-class extension may seem straightforward, but we also feel that this is not a weakness. Indeed, it is just the starting point for further analysis of the Bayes error rate (BER) estimator. Moreover, one needs to formally prove the theoretical properties of this estimator (Theorem 1) before exploring more advanced estimators (e.g., with denoising or robust version).
>
> Weakness 2. The motivation to consider features that are discrete mainly stems from the fact that in practice many classification problems utilize such features (or quantized values from real-valued features). As we pointed out in the footnote on page 2, one can easily extend several of our results on discrete features to real-valued ones (see also our response to Major Comment 3). For example, as the Reviewer commented, the estimator in Definition 6, which extends the one in eq.(11), works well for real-valued (as well as for discrete or mixture valued) features.
>
> Weakness 3. In our problem setting, the soft labels are available and they may be corrupted by some noise. In particular, the soft labels contain the value of $P_{Y|X}$ with noise; our proposed approach reduces the noise effect in the labels, instead of estimating the data distribution. This was the main reason of the sentence. We feel that, although the denoising method in Definition 6 applied to one-hot labels is equivalent to estimating the conditional probability, our approach proposes a novel perspective which could lead to new directions to effectively estimate the BER. Denoising methods have been largely studied and applied to several signal processing problems; this paper shows that they are effective also for BER estimation.
>
> Weakness 4. We believe that the considered SOTA BER bounds, namely the GHP and the $k$-NN error, leverage effective methods for estimating either the conditional probability or the divergence between the class conditional distributions. In this sense, our evaluations show that we outperform SOTA methods. That said, we agree with the Reviewer that a comparison with an additional approach that uses a SOTA technique for estimating the conditional probability, and then uses this to evaluate the BER would validate more our statement. We will add this comparison in the revised version.
>
> Major Comment 1. The parameter $r$ is a hyperparameter that should depend on the data feature space and on the underlying classification problem. Thus, it is not possible to choose an optimal $r$ before we look at the data and problem, and finding an optimal $r$ is a difficult task in general. In our experiments, we did an empirical search of the value of $r$ in a heuristic way. We chose the value of $r$ at the point where the estimated BER becomes stable (see Figure 9 in the supplementary material). The heuristic method that we used to find $r$ is similar to the elbow method for finding the number of clusters in $k$-means clustering algorithms.
>
> Major Comment 2. The main motivation for proposing our method lies in exploring new types of labels, i.e., the soft labels, which are starting to be utilized in many applications. By means of Theorem 1, using soft-valued labels with noise, we can estimate the BER by reducing the noise effect instead of estimating the data distribution, which is the approach pointed out by the Reviewer. We believe that for "many other classification methods performing well" the Reviewer intends methods based on estimating the class posterior probability. The main reason why we did not focus on such estimators is due to the hardness of estimating the data distribution if the data is complex or has high-dimensionality. Instead, our estimator just uses a cluster-like method that is applicable to most dataset (see also Remark 2). In the revised version, we will add such discussion.
>
> Major Comment 3. We believe that the assumption of having features that are discrete is somehow without loss of generality in practice. Indeed, several classification problems utilize discrete features, or quantized values from real-valued features. This is the main reason for our assumption. That said, we highlight that Theorem 1 and Theorem 2 also hold for the infinite feature space. Theorem 3 is the only result that we should carefully analyze for the case of infinite feature space. However, we feel that this is doable. In the revised version, we will add this.
>
> Minor Comment 1. Any $L$-dimensional ellipsoid (after rotation) can be defined as $\frac{x_1^2}{a_1^2} + \cdots + \frac{x_L^2}{a_L^2} = 1$. The $L$ radii (and not radiuses) are $\{a_1, a_2, \cdots, a_L\}$. In the $2$-dimensional space, one can imagine that an ellipsoid has $2$ main axes defining it, and the radii are the half length of each main axes in the ellipsoid.
>
> Minor Comment 2. Yes, a larger $K$ gives more robustness; $K=\ln n$ and $K=\sqrt{n}$ should be interchanged. We will fix this in the revised version.
>
> Minor Comment 3. We believe that the assumption of zero mean noise is without loss of generality (wlog). If a noise distribution has a non-zero mean, we can subtract the mean of the noise from the noise distribution, which then becomes zero mean. However, the assumption of i.i.d. is not wlog; we feel that this may be the source of confusion for the Reviewer. In the revised version, we will clarify that only the assumption of zero mean is wlog.
>
> Minor Comment 4. We will fix the definition of a $\gamma^2$-sub-Gaussian random variable, i.e., a random variable $X$ satisfying $\mathbb{E}[e^{\lambda (X-\mathbb{E}[X])}] \leq e^{\frac{\lambda^2 \gamma^2}{2}},~\forall \lambda \in\mathbb{R}$.
>
> Minor Comment 5. We will refer to it as Nadaraya-Watson estimator with the Parzen window kernel.
>
> Minor Comment 6. We will replace $P_e(\cdot)$ with $\mathcal{E}(\cdot)$.

---

> > ### Comment · Reviewer_2nax · 2023-08-16
> > **Re: Rebuttal by Authors**
> >
> > I thank the authors for the response. I have a few follow-up questions.
> >
> > > Weakness 2. The motivation to consider features that are discrete mainly stems from the fact that in practice many classification problems utilize such features (or quantized values from real-valued features). As we pointed out in the footnote on page 2, one can easily extend several of our results on discrete features to real-valued ones (see also our response to Major Comment 3). For example, as the Reviewer commented, the estimator in Definition 6, which extends the one in eq.(11), works well for real-valued (as well as for discrete or mixture valued) features.
> >
> > There are also many problems with continuous features, especially in modern applications. I would think it as a limitation.
> >
> > > Major Comment 2 (...) the Reviewer intends methods based on estimating the class posterior probability.
> >
> > Do we really need an estimate of class posterior probabilities? If we only need the argmax of them (like idx(.) does), we may only need a classifier.
> >
> > > Minor Comment 3. We believe that the assumption of zero mean noise is without loss of generality (wlog). If a noise distribution has a non-zero mean, we can subtract the mean of the noise from the noise distribution, which then becomes zero mean. However, the assumption of i.i.d. is not wlog; we feel that this may be the source of confusion for the Reviewer. In the revised version, we will clarify that only the assumption of zero mean is wlog.
> >
> > If the mean of the noise is unknown, one has to estimate it for centering. Can we still make the zero-mean assumption for free?

---

> > > ### Author Response · Authors · 2023-08-16
> > >
> > > We thank the Reviewer for their thorough reading of our rebuttal, and for the additional follow-up questions. We hope that our response properly addresses the Reviewer’s comments.
> > >
> > > ### Regarding Weakness 2.
> > > We totally agree with the Reviewer that continuous features are very relevant, and we apologize if our response was perceived as understating their importance. While some of our results (Theorem 1 and Theorem 2) can be easily shown to hold also for the infinite feature space, the estimator in eq.(11) assumes that the set of the features is finite. As the Reviewer also pointed out, this is a limitation of this estimator, and was one of the main motivations to propose the cluster-based estimator in eq.(15). This estimator indeed extends the one in eq.(11) and it works well for real-valued (as well as for discrete or mixture valued) features. The next step will consist of proving a similar result as in Theorem 3 for the estimator in eq.(15). We are currently investigating this problem, and we are hopeful that this is doable. We will also add this discussion in the Discussion/Conclusion section that we will include in the revised version of the paper.
> > >
> > > ### Regarding Major Comment 2.
> > > Our approach in Definition 5 consists of two phases: we first propose a denoising label estimator in eq.(11) – which as the Reviewer pointed out estimates the class posterior probabilities – and then we use this to estimate the BER – by using eq.(12). In other words, we are not using a classifier to estimate the BER. We may have missed something in the Reviewer’s concern and, if this is the case, we would appreciate further elaboration on which existing classification method(s) can be used to estimate the BER.
> > >
> > > ### Regarding Minor Comment 3.
> > > Yes, the Reviewer is absolutely correct in saying that, if the mean is not known, then it has to be estimated. The wording ‘without loss of generality’ conveys the fact that estimating the mean of the noise (which is beyond the scope of our paper) is often regarded as a trivial task in many applications when the mean is unknown. However, we acknowledge that this might cause confusion, and we will make sure to remove the ‘without loss of generality’ in the revised version of the paper.

---

> > > > ### Comment · Reviewer_2nax · 2023-08-21
> > > > **Re: Regarding Major Comment 2**
> > > >
> > > > > Our approach in Definition 5 consists of two phases: we first propose a denoising label estimator in eq.(11) – which as the Reviewer pointed out estimates the class posterior probabilities – and then we use this to estimate the BER – by using eq.(12). In other words, we are not using a classifier to estimate the BER. We may have missed something in the Reviewer’s concern and, if this is the case, we would appreciate further elaboration on which existing classification method(s) can be used to estimate the BER.
> > > >
> > > > The BER estimator in Eq. (12) uses the denoised label $\boldsymbol{s}(\boldsymbol{x}_i)$ only through $\operatorname{idx}(\boldsymbol{x}_i)$. In the definition of $\operatorname{idx}(\cdot)$, we take the argmax of the scores $(\boldsymbol{s}(\boldsymbol{x}_i))_j$, $j \in [M]$. This is essentially performing classification, and we can replace the whole part with any good classifier if I'm not mistaken.

---

> > > > > ### Author Response · Authors · 2023-08-21
> > > > >
> > > > > Thank you very much for a positive suggestion for improvement.
> > > > > Yes, it can be replaced by any classifier.
> > > > > We didn’t try to find it, but we think that our classifier is almost optimal under zero mean noise and discrete feature assumption. If one adds more assumptions on the noise distribution, there could be a better classifier.

---

### Official Review · Reviewer_7qAC · 2023-07-06

**Soundness:** 2 fair
**Presentation:** 2 fair
**Contribution:** 2 fair
**Rating:** 5
**Confidence:** 4

**Summary:**

This paper aims to design a tighter estimate for the Multi-class classification error. The paper analyzes several theoretical aspects of the suggested Bias Error rate  estimator, including its consistency, unbiasedness, convergence rate, variance, and robustness. Moreover the authors utilize a denoising method to reduce label noise if such is present and improve robustness to the outliers by incorporating the median-of-means estimator. They validate the effectiveness of our theoretical results via experiments both on synthetic data under various noise settings and on real data (CIFAR-10H, Fashion-MNIST-H and MovieLens -- movie recommendation dataset)


**Strengths:**

The paper raises an important issue of estimating performance limit of the multi-class classifier.  It provides thorough theoretical analysis, backed up by some experimentation.

**Weaknesses:**

I'm struggling to see an impact of this paper. It aims to demistify optimal performance of multi-class classification, but I did not see any illustration of this in the paper. I would expect to see performance of SOTA method on a complex task (e.g., ImageNet) vs. the provided estimator, but the paper uses relatively easy classification tasks for illustration of the bounds. So not clear what is the applicability of this paper for more challenging tasks of Computer Vision and NLP. Also I would expect some theoretical derivation on what is a good feature space for which the suggested denoising holds. Note that the denoising itself is not novel, and seems a bit ad-hoc solution to improve the accuracy of the estimator.

**Questions:**

I would love the authors to explain what is the impact of their work and experiment (maybe in Discussion/ Conclusion section that is currently missing).

**Limitations:**

My main concern is that the paper does not have a real impact on more complex Deep Learning tasks of NLP and Computer Vision.

---

> ### Author Rebuttal · Authors · 2023-08-10
>
> We thank the Reviewer for their time and effort put into thoroughly reading and reviewing our manuscript.
>
> Regarding the impact/applicability of our paper. We would like to first note that we did consider a complex classification task for the evaluation of our bounds using the MovieLens dataset. This is indeed a challenging task in computer vision where a multi-modal network model classifies a two-hour movie (input feature) with audio. None of the SOTA Bayes error rate (BER) estimators would be applicable to such a complex task since they either require to find the data distribution/divergence or have a prohibitive complexity. Differently, our estimator simply requires labels and a denoising method. Second, we would like to point out that the reason we evaluated the bounds for some easy classification tasks (e.g., Gaussian noise) was to showcase the effectiveness of them in estimating the BER in scenarios where this is indeed known. In summary, we agree that  including more experiments on complex tasks would strengthen the paper. However, we also feel that this would go beyond the scope of this single paper.
> As also suggested by another Reviewer, in the revised version of the paper, we will add a Conclusion section with such a discussion.
>
> Regarding the denoising method. The results presented in this paper show that the proposed denoising method is consistent for any (finite) feature space in the asymptotic (see Theorem 3). We agree that finding a good feature space could be the next natural step, and we thank the Reviewer for this comment. In particular, one could analyze the rate of convergence of our BER estimator paired with the denoising method as a function of the characteristics (e.g., cardinality, distribution) of the feature space. We will add this discussion in the Conclusion section in the revised version of the paper. Finally, we would like to note that our denoising technique is theoretically universal (see Theorem~3) and hence, we feel that it is not an ad-hoc solution. Of course, if we have knowledge about the noise distribution in the dataset, then we can improve the accuracy of the estimator, which might be indeed an ad-hoc solution. However,  we here make a very general assumption on the noise distribution (i.e., i.i.d. and zero-mean). In summary, we agree that the denoising technique can be further theoretically analyzed, but we do not feel that this represents a weakness of our work.

---

> > ### Comment · Reviewer_7qAC · 2023-08-16
> > **Follow up on rebuttal**
> >
> > I would like to thank the authors and other reviewers for the discussion on this page.
> >
> > I do agree with the Reviewer 2nax that the denoising method is disaappoitning. I believe there should be a thorough discussion about the limitations of the proposed approach. This goes back to my comment on impact. However, I also agree with other reviewers that in overall the paper might be interesting to the community and encourage further research.
> >
> > Given the feedback from other reviewers and thoughtful follow-ups of the authors, I changed my feedback to 5.

---

> > > ### Author Response · Authors · 2023-08-18
> > >
> > > We thank the Reviewer for providing further feedback on our paper and raising the score to 5. We acknowledge that the denoising method presented in eq.(11) is specifically suitable for discrete features, which does represent a limitation of our work. However, it is important to highlight that we have shown to be aware of this limitation by introducing a cluster-based denoising approach (eq.(15) in Definition 6), designed to be effective with any type of feature. We are grateful to the reviewer for pointing us out to the absence of a discussion addressing this limitation. We will include a detailed discussion in the revised version of the paper.

---

### Decision · Program_Chairs · 2023-09-21

**Decision:**

Accept (poster)

**Comment:**

This seems to be classical "borderline" paper, and all reviewers (more or less) share this opinion. There are several interesting aspects, like the novel robust estimator proposed that comes with some theoretical performance guarantees. On the other hand, there are also several more critical remarks, such as lacking clarity regarding the "true" depth and impact of this work.
There was quite some discussions among reviewers and authors, and during these discussions several critical points raised by the reviewers could be addressed, at least up to a point where it is likely that an improved final version of this paper could be accepted.
Therefore, I recommend acceptance, but at the same time I would like to encourage the authors to improve several parts in the final version of the paper according to the discussions.